# DYSLEXIFY: A MECHANISTIC DEFENSE AGAINST TYPOGRAPHIC ATTACKS IN CLIP

**Lorenz Hufe** [1,2], **Constantin Venhoff** [2], **Erblina Purelku** [1],
**Maximilian Dreyer** [1], **Sebastian Lapuschkin** [1,3], **Wojciech Samek** [1,4]
[1] Fraunhofer Heinrich Hertz Institute, [2] University of Oxford
[3] Technological University Dublin, [4] Technische Universität Berlin
lorenz.hufe@hhi.fraunhofer.de

## ABSTRACT

Typographic attacks exploit multi-modal systems by injecting text into images, leading to targeted misclassifications, malicious content generation and even Vision-Language Model jailbreaks. In this work, we analyze how CLIP vision encoders behave under typographic attacks, locating specialized attention heads in the latter half of the model's layers that causally extract and transmit typographic information to the cls token. Building on these insights, we introduce Dyslexify – a method to defend CLIP models against typographic attacks by selectively ablating a typographic circuit, consisting of attention heads. Without requiring finetuning, Dyslexify improves performance by up to 22.06% on a typographic variant of ImageNet-100, while reducing standard ImageNet-100 accuracy by less than 1%, and demonstrate its utility in a medical foundation model for skin lesion diagnosis. Notably, our gradient-free approach remains competitive with current state-of-the-art typographic defenses that rely on finetuning. To this end, we release a family of dyslexic CLIP models which are significantly more robust against typographic attacks. These models serve as suitable drop-in replacements for a broad range of safety-critical applications, where the risks of text-based manipulation outweigh the utility of text recognition.

Poject page: https://hufe.info/dyslexify/
Code: https://github.com/lowlorenz/dyslexify

## 1 INTRODUCTION

CLIP models are increasingly adopted as general-purpose vision–language representations, enabling applications in zero-shot classification, retrieval, diffusion-based generative models, and large-scale vision–language models (VLMs). Their versatility has further driven adoption in safety-relevant domains such as healthcare (Yang et al., 2024; Wang et al., 2022; Eslami et al., 2023), remote sensing (Liu et al., 2024a; Vivanco et al., 2023; Li et al., 2023), and content moderation (Schuhmann et al., 2022; Liu et al., 2025; Reyes et al., 2025). However, despite their widespread use, CLIP models remain vulnerable to typographic attacks: inserting text into an image can mislead classification, trigger malicious generations, or even jailbreak multi-modal systems (see Fig. 1).

Existing defenses against typographic attacks require gradient-based optimization. While effective to some extent, these methods require substantial computational resources and lack interpretability into the mechanisms underlying CLIP's behavior.

In this work, we introduce Dyslexify a gradient-free defense that directly targets model circuits responsible for the vulnerability to typographic attacks. By identifying and ablating a set of attention heads with demonstrable causal effects, we construct dyslexic CLIP models that are substantially more robust to typographic attacks. Our method scales seamlessly to billion-parameter models, making it applicable to state-of-the-art multi-modal systems. Beyond improving robustness, our approach also enhances interpretability of CLIP models, enabling targeted intervention that are computationally efficient and easily integrated into existing pipelines without additional overhead.

The contributions of this work are:

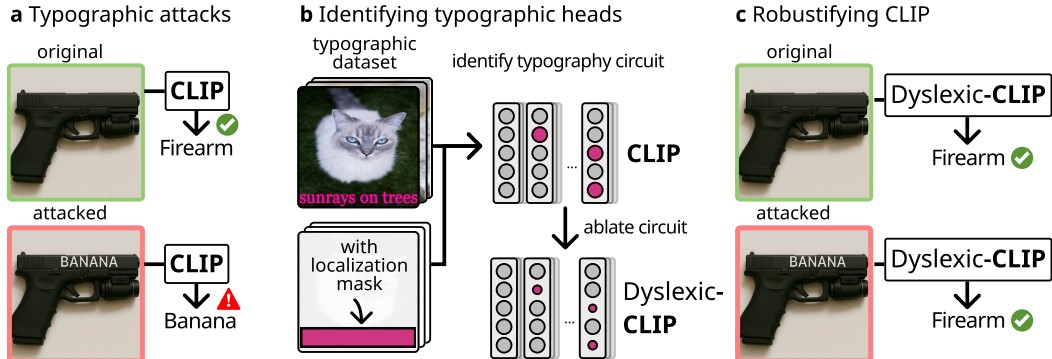

Figure 1: Defending CLIP against typographic attacks with Dyslexify a) Adversarial text in images can dominate CLIP's representation and lead to misclassification. b) We construct a circuit of attention heads responsible for transmitting typographic information. c) By suppresses the typographic circuit, we defend against typographic attacks without a single gradient step.

- **Mechanistic Understanding:** We present the Typographic Attention Score to locate specialized typographic attention heads and demonstrate their causal role in typographic attacks within CLIP models through controlled interventions.

- **Gradient-Free Defense:** We introduce a method that utilizes circuit ablation to effectively defend against typographic attacks, while maintaining general visual capabilities. Due to its gradient-free nature, Dyslexify seamlessly scales to billion-parameter models on consumer grade hardware.

- **Empirical Validation:** We validate Dyslexify across a diverse set of zero-shot classification tasks, demonstrating that our approach improves robustness to typographic attacks by up to 22.06% on a typographic version of Imagenet-100 while maintaining high accuracy on non-typographic benchmarks.

- **Medical Use Case:** We show that typographic attacks pose a tangible risk to safety-critical medical foundation models, and demonstrate that Dyslexify can substantially mitigate this vulnerability.

- **Model Release:** To facilitate safer deployment, we release a family of dyslexic CLIP models with reduced typographic sensitivity, suitable for use in safety-critical applications.

Our approach provides a practical, interpretable, and computationally efficient typographic defense, paving the way for safer multimodal systems without the need for fine-tuning.

## 2 RELATED WORK

CLIP models (Radford et al., 2021) are pretrained on large-scale image–text datasets such as Laion-5b (Schuhmann et al., 2022), aligning global image features with textual descriptions for strong zero-shot transfer. This reliance on textual supervision also makes them vulnerable to typographic attacks.

Typographic attacks (Goh et al., 2021) insert written text into an image to maliciously alter a model's behavior. Recent work demonstrates that typographic attacks can degrade model performance, bypass safety filters (jailbreaking) and hijack goals in Vision Language Models (VLMs) (Qraitem et al., 2024; Kimura et al., 2024; Gong et al., 2025; Cao et al., 2024; Westerhoff et al., 2025), trigger harmful content generation in image-to-image pipelines (Cheng et al., 2024), and cause targeted misclassification in object detection and zero-shot classification settings (Materzyńska et al., 2022; Ilharco et al., 2022; Azuma & Matsui, 2023; Westerhoff et al., 2025; Dreyer et al., 2025).

Several defenses have been proposed, they rely either on fine-tuning the model (Ilharco et al., 2022), learning a projection matrix (Materzyńska et al., 2022), incorporating a learnable text-token called Defense-Prefix (Azuma & Matsui, 2023), or employing Sparse Autoencoders (Joseph et al., 2025). Crucially, none of these approaches offer a gadient-free method for mitigating typographic attacks.

In contrast, our work introduces a controllable intervention at inference time by directly locating and suppressing the components responsible for typographic sensitivity, without requiring gradients or fine-tuning.

## 3 MOTIVATION: LOCATING LAYERS OF TYPOGRAPHIC UNDERSTANDING

To better understand CLIP's vulnerability to typographic attacks and to motivate our method, we begin by investigating which layers and components are responsible for typographic understanding using linear probes.

**Typographic datasets:** We further construct typographic attack datasets from standard image classification datasets $D = \{(x_i, y_i)\}_{i=1}^n$ by assigning each input example $x_i$ an additional typographic label $z_i \neq y_i$ different from the original label $y_i$, and overlaying a corresponding textual description of $z_i$ onto the original image $x_i$ at a random location as shown in Fig. 2a. We denote these modified datasets with the suffix "-typo". More information on the datasets is provided Appendix F.

Extending the ImageNet-100 dataset into the ImageNet-100-typo dataset, we can evaluate typographic understanding by training linear probes on the cls token embedding at each layer of OpenCLIP models, ranging in scale from ViT-B to ViT-bigG (Ilharco et al., 2021).

Formally, for a layer $\ell$ we define a linear probe $P_\ell$

$$\hat{y}_\ell(x) = w^\top h_{\text{cls}}^\ell + b \,, \tag{1}$$

where $h_{\text{cls}}^\ell \in \mathbb{R}^d$ denotes the activation of the model at layer $\ell$ on the cls token for an input sample $x$, and $w$ and $b$ are the probe's weight vector and bias term, respectively. We train two types of probes: $P_{\text{img},\ell}$, which predicts the object label $y$ and $P_{\text{typo},\ell}$, which predicts the typographic label $z$. The accuracy of a probe $P$ is denoted by $\text{Acc}(P)$.

Fig. 2b shows that $\text{Acc}(P_{\text{typo},\ell}) > 0.99$ at the final layer for all tested models, indicating that CLIP models can distinguish the typographic classes. Furthermore it shows, that $\text{Acc}(P_{\text{typo},\ell})$ is low in early layers, but exhibit a sharp increase in the latter half of the model.

Fig. 2c shows that $\text{Acc}(P_{\text{img},\ell})$ improves gradually over the layers, while $\text{Acc}(P_{\text{typo},\ell})$ exhibits a sharp performance rise around second half of the models layers.

Fig. 2d highlights the effect of the attention and the MLP blocks onto $\text{Acc}(P_{\text{img},\ell})$ and $\text{Acc}(P_{\text{typo},\ell})$. Attention layers consistently improve accuracy indicating that they add linearly decodable information to the cls token. In contrast, the MLP layers tend to reduce accuracy. We show in Appendix B, that the Intrinsic Dimensionality (ID) of the signal decreases after the MLP blocks, suggesting that the MLP compresses or discards information.

## 4 DYSLEXIFY: A DEFENSE AGAINST TYPOGRAPHIC ATTACKS

To defend CLIP against typographic attacks we present Dyslexify: a framework for detecting and suppressing typographic circuits. Based on the finding in Section 3 that attention heads are responsible for adding typographic information, we only consider attention heads for the circuit construction. We define a circuit as a subset $\mathcal{C} \subseteq \Psi$, where $\Psi$ denotes the set of attention heads in CLIP

$$\Psi = \{\mathcal{H}_{i,\ell} \mid i \in \{0, \dots, I\}, \ \ell \in \{0, \dots, L\}\} \tag{2}$$

with $I$ heads per layer and $L$ layers in total, where $\mathcal{H}_{i,\ell}$ denote the $i$th attention head in layer $\ell$.

To robustify a model $\mathcal{M}$ against typographic attacks, we conduct *circuit-ablation* of a typographic circuit $\mathcal{C}$. Circuit-ablation modifies here only the residual stream of the cls token, leaving all other computations intact. Specifically, the residual update of the cls token is given by

$$z_{\text{cls}}^\ell = h_{\text{cls}}^\ell + \text{MLP}(h_{\text{cls}}^\ell) \tag{3}$$

$$h_{\text{cls}}^{\ell+1} = z_{\text{cls}}^\ell + \sum_{i=1}^{H} \mathcal{H}_{i,\ell,\text{cls}}(z_{\text{cls}}^\ell) \tag{4}$$

where $z_{\text{cls}}^\ell$ is the cls activation after the layer-$\ell$ MLP block is applied to $h_{\text{cls}}^\ell$, and $\mathcal{H}_{i,\ell,\text{cls}}(z_{\text{cls}}^\ell)$ is the contribution of head $\mathcal{H}_{i,\ell}$ to the cls token.

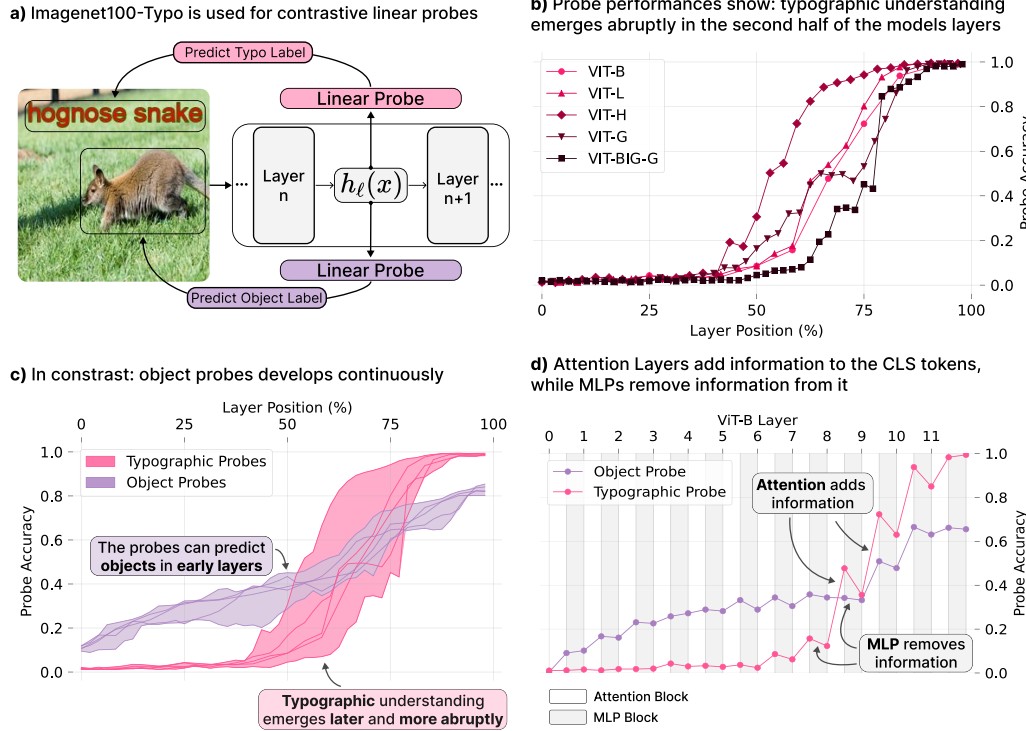

Figure 2: Investigating where typographic understanding emerges in CLIP. **a)** We train two linear probes on all layers of CLIP models. Probe $P_{\text{img},\ell}$ is used to predict the text label of each sample while $P_{\text{typo},\ell}$ is trained to predict the typographic class. **b)** $P_{\text{typo},\ell}$ shows a consistent pattern across all model sizes: typographic information emerges abruptly in the second half of the models layers. **c)** This trend is not true for the object probes $P_{\text{img},\ell}$. Object specific information builds gradually over the layers. Each line in the shaded area represents one CLIP model. **d)** While attention layers seem to add linearly decodable information to the `cls` token, MLP layers remove or obscure information.

We define ablation of a typographic circuit $\mathcal{C}$ as

$$\mathcal{H}_{i,\ell,\text{cls}}(z_{\text{cls}}^{\ell}) \leftarrow 0 \quad \text{for all } \mathcal{H}_{i,\ell} \in \mathcal{C}, \tag{5}$$

while leaving all spatial contributions unchanged. We use $\mathcal{M}_{\mathcal{C}}$ to denote a model $\mathcal{M}$ in which circuit $\mathcal{C}$ is ablated.

## 4.1 TYPOGRAPHIC ATTENTION SCORE

Building on Hung et al. (2024) we introduce the Typographic Attention Score $T_{i,\ell}$ to guide the circuit construction. Intuitively the score measures the amount of spatial attention a head $\mathcal{H}_{i,\ell}$ dedicates to typographic content.

Given a head $\mathcal{H}_{i,\ell}$ and an input $x$, we write $A_{i,\ell}(x) \in [0,1]^{T+1}$ to denote the `cls` tokens attention pattern, where $T$ is the number of spatial tokens and the additional entry corresponds to the `cls` token. We further define the spatial `cls`-attention pattern $A_{i,\ell}^{*}(x) \in [0,1]^{T}$ which excludes the `cls`-to-`cls` entry $A_{i,\ell,\text{cls}}(x)$, such that

$$A_{i,\ell}(x) = \big(A_{i,\ell,\text{cls}}(x), \ A_{i,\ell}^{*}(x)\big). \tag{6}$$

We write $A_{i,\ell,t}^{*}(x)$ to index the $t$th element of the pattern. Formally the score is given by:

$$T_{i,\ell} = \sum_{x \in \mathcal{D}} \frac{\sum_{t=1}^{T} \mathbb{1}(t)\, A_{i,\ell,t}^{*}(x)}{\sum_{t=1}^{T} A_{i,\ell,t}^{*}(x)}. \tag{7}$$

Where $x$ is a data point, $\mathcal{D}$ is the dataset, and $\mathbb{1}$ is an indicator function so that $\mathbb{1}(t) = 1$ if the input patch associated with the token at index $t$ corresponds to typographic content and is zero otherwise.

## 4.2 TYPOGRAPHIC CIRCUIT CONSTRUCTION

To defend against typographic attacks without degrading zero-shot classification, Dyslexify constructs a typographic circuit $\mathcal{C}$. The circuit is built iteratively while monitoring the accuracy of the circuit-ablated model $\mathcal{M}_{\mathcal{C}}$ on a non-typographic benchmark $D_{\text{img}}$ and a typographic benchmark $D_{\text{typo}}$, ensuring that the accuracy on $D_{\text{img}}$ never decreases by more than a threshold $\epsilon \in \mathbb{R}$.

Our procedure consists of two steps: (i) rank all attention heads $\mathcal{H}_{i,\ell}$ by their typographic score $T_{i,\ell}$; (ii) add heads to $\mathcal{C}$ in descending order of $T_{i,\ell}$, evaluating accuracy after each addition.

Let $\text{Acc}(\mathcal{M}, D)$ denote the accuracy of model $\mathcal{M}$ on dataset $D$. For each candidate head $\mathcal{H}$, we compute

$$\Delta\text{Acc}_{\text{img}} = \text{Acc}(\mathcal{M}, D_{\text{img}}) - \text{Acc}(\mathcal{M}_{\mathcal{C}\cup\mathcal{H}}, D_{\text{img}}), \tag{8}$$

$$\Delta\text{Acc}_{\text{typo}} = \text{Acc}(\mathcal{M}_{\mathcal{C}\cup\mathcal{H}}, D_{\text{typo}}) - \text{Acc}(\mathcal{M}_{\mathcal{C}}, D_{\text{typo}}), \tag{9}$$

where $\Delta\text{Acc}_{\text{img}}$ measures the accuracy drop on $D_{\text{img}}$ relative to the base model, and $\Delta\text{Acc}_{\text{typo}}$ measures the incremental gain on $D_{\text{typo}}$ from adding head $\mathcal{H}$ to the current circuit $\mathcal{C}$.

If $\Delta\text{Acc}_{\text{typo}} \leq 0$, the head $\mathcal{H}$ is skipped, as it does not improve robustness to typographic attacks. If the head is not skipped and $\Delta\text{Acc}_{\text{img}} < \epsilon$, the head is added to $\mathcal{C}$; otherwise, the algorithm terminates. In addition, if more than $k \in \mathbb{N}$ heads are skipped consecutively, the algorithm also terminates.

We refer to the final model $\mathcal{M}_{\mathcal{C}}$, equipped with the constructed circuit $\mathcal{C}$, as the *dyslexic model*.

---

**Algorithm 1** Dyslexify

---

1: Initialize circuit $\mathcal{C} \leftarrow \emptyset$, skip counter $s \leftarrow 0$.
2: Set hyperparameters: tolerance $\epsilon$, max skips $k$.
3: Rank heads $\mathcal{H}_{i,\ell}$ by score $T_{i,\ell}$.
4: **for** head $\mathcal{H}_{i,\ell}$ in descending order of $T_{i,\ell}$ **do**
5:     Compute $\Delta\text{Acc}_{\text{img}}, \Delta\text{Acc}_{\text{typo}}$.
6:     **if** $\Delta\text{Acc}_{\text{typo}} \leq 0$ **then**
7:         $s \leftarrow s + 1$;
8:         **if** $s \geq k$ **then break; end if**
9:         **continue**;
10:     **end if**
11:     **if** $\Delta\text{Acc}_{\text{img}} \geq \epsilon$ **then break; end if**
12:     Add $\mathcal{H}_{i,\ell}$ to $\mathcal{C}$; $s \leftarrow 0$.
13: **end for**
14: Return final circuit $\mathcal{C}$.

---

## 5 EXPERIMENTS

### 5.1 EVALUATING THE TYPOGRAPHIC ATTENTION SCORE

We construct localized typographic dataset consisting of 10,000 natural images from Unsplash[1], originally containing minimal typographic content. To efficiently analyze the attention patterns, we synthetically introduce typographic content at the bottom center of the image, responding to the lowest two token rows in the spatial grid.

For each attention head $\mathcal{H}_{i,\ell}$ we extract the attention pattern $A_{i,\ell}^{*}(x)$ over this localized dataset. We use the known spatial bias to define the indicator mask $\mathbb{1}(t) \in \{0,1\}$, where $\mathbb{1}(t) = 1$ at the the typographic region and $\mathbb{1}(t) = 0$ elsewhere.

Fig. 3 shows the resulting Typographic Attention Scores $T_{i,\ell}$ for ViT-B. A small subset of heads shows high scores of up to $T_{i,\ell} \geq \mu(T) + 2\sigma(T)$, revealing a strong spatial bias towards typography.

---

[1] https://huggingface.co/datasets/wtcherr/unsplash

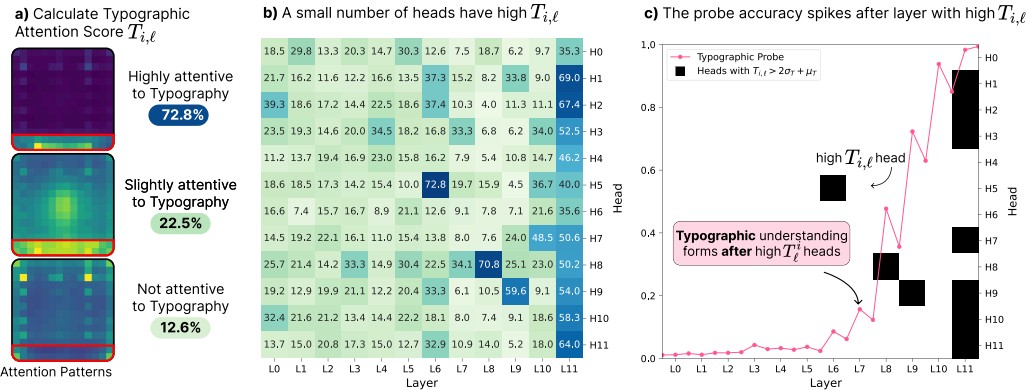

Figure 3: Analysis of the Typographic Attention Score. **a)** For each head in the model we calculate the Typographic Attention Score $T_{i,\ell}$, utilizing the spatial bias in the Unsplash-typo dataset. **b)** Depiction of ViT-B's $T_{i,\ell}$ scores. While most attention heads do not show any spatial bias in their attention patterns, a few attention heads indicate significantly elevated scores, exceeding $T_{i,\ell} \geq \mu(T) + 2\sigma(T)$. Those heads only occur in the second half of the models layers **c)** Overlaying the linear probes with significantly elevated $T_{i,\ell}$ scores , highlights an interesting correlation. Only after the attention heads with exceptionally high $T_{i,\ell}$ scores are passed the model the accuracy of $P_{\text{typo},\ell}$ begins to increase rapidly.

Furthermore we observe, that the spike in $\text{Acc}(P_{\text{typo},\ell})$ only occurs after layers with high $T_{i,\ell}$ heads. More results can be found in Appendix K.

## 5.2 EVALUATING THE TYPOGRAPHIC CIRCUITS

To construct the circuits $\mathcal{C}$, we set the tolerance to $\epsilon = 0.01$, set the maximum number of consecutive skips to $k = 10$, use the ImageNet-100 training split as $D_{\text{img}}$, and ImageNet-100-typo as $D_{\text{typo}}$. Table 5 shows that the resulting circuits are sparse, covering at most 10.1% of $\Psi$.

Fig. 4 plots $\text{Acc}(\mathcal{M}_{\mathcal{C}}, D_{\text{img}})$ and $\text{Acc}(\mathcal{M}_{\mathcal{C}}, D_{\text{typo}})$ as heads are added. Dyslexify, improves accuracy on ImageNet-100-typo's train set by more than 20% across all evaluated models, while limiting the drop in ImageNet-100 accuracy to below 1%. More results can be found in Fig. 9.

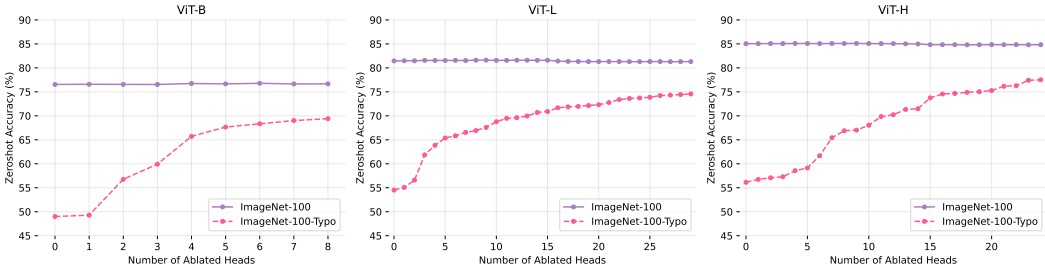

Figure 4: Tradeoff between general accuracy and typographic robustness as a function of the number of ablated heads. Ablations are applied in decreasing order of $T_{i,\ell}$.

### 5.2.1 DEMONSTRATING CAUSALITY OF TYPOGRAPHIC CIRCUITS

We observe that attention heads in $\mathcal{C}$ utilize their `cls` self-attention as attention sinks (Xiao et al., 2023) depending on the presence of typography in the image. Further details are provided in Appendix C. Building on those findings we showcase the casual nature of these attention heads in this section.

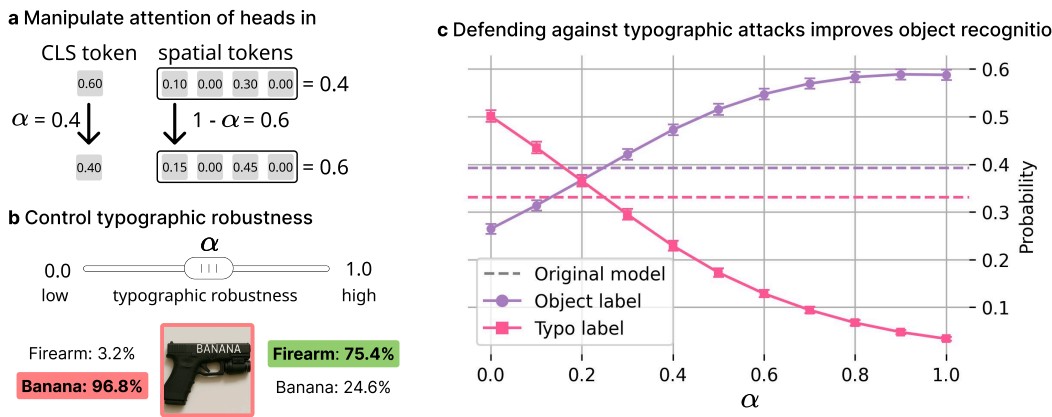

Figure 5: Controlling typographic vulnerability by manipulating attention sinks in circuit heads. a) We set the `cls` token attention to $\alpha$ and rescale the spatial token attentions to sum to $1 - \alpha$. b) Increasing $\alpha$ raises attention to spatial tokens, amplifying typographic understanding. c) Decreasing $\alpha$ increases typographic robustness, increasing the probability of predicting the true object class.

To demonstrate the causal role of these heads in typographic vulnerability, we manipulate their attention patterns. Specifically, we construct

$$A_{i,\ell}^{\alpha} = \left[\alpha, \ A_{i,\ell}^* \cdot (1-\alpha)/\|A_{i,\ell}^*\|\right], \tag{10}$$

where $\alpha \in \{0.0, 0.1, \ldots, 0.9, 1.0\}$. The scaling factor $(1-\alpha)/\|A_{i,\ell}^*\|$ ensures that the attention distribution remains normalized.

We then evaluate the effectiveness of typographic attacks under these manipulations by tracing the predicted label probabilities $p(y_{\text{text}})$ and $p(y_{\text{typo}})$ as a function of $\alpha$ on the ImageNet-100-typo dataset.

**Results:** Fig. 5 shows that increasing $\alpha$ causally reduces the effectiveness of typographic attacks, while decreasing $\alpha$ amplifies it. This provides direct causal evidence: as circuit heads allocate more weight to the `cls` sink, the attack signal is suppressed; conversely, when $\alpha$ is low and spatial attention dominates, $p(y_{\text{typo}})$ increases, indicating that typographic information is transferred from spatial tokens into the `cls` representation.

### 5.3 EVALUATING THE DYSLEXIC MODELS

To evaluate Dyslexify we construct dyslexic OpenClip model variants: ViT-B, L, H, G, and BigG and conduct zero-shot classification experiments. Concretely we first evaluate the effectiveness of our defense against typographic attacks and secondly measure the zero-shot object classification capabilities on non-typographic datasets. For each model, we record the accuracy difference between the original model and dyslexic model. A detailed description of the datasets is given in Appendix F.

**Results:** Table 1 shows that Dyslexify yields consistent robustness improvements across both real-world typographic attack datasets and synthetic benchmarks. The observed gains are substantial – up to +31% accuracy – and occur across all evaluated datasets, suggesting that the identified typographic circuits capture generalizable failure modes rather than dataset-specific artifacts.

Table 2 further demonstrates that Dyslexify preserves performance on standard vision datasets. In nearly all cases, deviations remain within $\pm 1\%$ of the base model, the only exception being ViT-L showing the largest decline ($-1.74\%$) on Aircraft and ($-1.17\%$) on Food-101, which is close to the tolerance bound $\epsilon = 1\%$. This indicates that Dyslexify achieves a favorable robustness–accuracy trade-off: substantial robustness gains are obtained while standard zero-shot performance is essentially maintained.

Table 1: Comparison of dyslexic model performance on datasets of typographic attacks across model sizes, showing accuracy changes relative to the base model, with improvements (↑) or declines (↓). IN denotes ImageNet, and the suffix -T indicates the corresponding typographic version of the dataset.

| Model | Real Typographic | | | Synthetic Typographic | | |
|---|---|---|---|---|---|---|
| | RTA-100 | Disentangling | Paint | IN-100-T | Food-101-T | Aircraft-T |
| B | 68.30↑12.00 | 85.00↑31.11 | 72.73↑14.55 | 66.84↑19.90 | 78.27↑22.64 | 16.23↑5.91 |
| L | 71.00↑16.60 | 60.56↑10.00 | 76.36↑14.55 | 72.22↑20.32 | 82.15↑26.55 | 23.34↑9.51 |
| H | 68.30↑15.20 | 72.22↑26.67 | 70.91↑21.82 | 75.34↑21.26 | 83.01↑28.68 | 29.40↑8.07 |
| G | 62.00↑12.00 | 67.22↑9.44 | 71.82↑16.36 | 68.76↑22.06 | 73.05↑20.21 | 27.69↑3.45 |
| Big-G | 72.90↑11.90 | 68.33↑20.00 | 69.09↑21.82 | 78.64↑16.74 | 84.69↑25.98 | 41.61↑16.29 |

Table 2: Comparison of dyslexic model performance on non-typographic datasets across model sizes, showing accuracy changes relative to the base model, with improvements (↑) or declines (↓).

| Model | Not Typographic | | |
|---|---|---|---|
| | Aircraft | Food-101 | ImageNet-100 |
| B | 27.72↓0.12 | 84.97↓0.99 | 75.00↑0.64 |
| L | 34.62↓1.74 | 89.31↓1.17 | 79.52↓0.24 |
| H | 43.98↑0.12 | 92.29↓0.24 | 83.40↓0.34 |
| G | 44.07↓0.30 | 91.47↓0.70 | 82.58↓0.66 |
| Big-G | 50.47↓0.39 | 92.55↓0.42 | 84.72↓0.34 |

## 5.4 COMPARING TO BASELINES

We compare Dyslexify to Defense-Prefix (DP) (Azuma & Matsui, 2023), which introduces a learnable prefix token for CLIP's language transformer on OpenCLIP ViT-L without fine-tuning the full ViT. Following their setup, we train the DP on the ImageNet-100-typo training split with a learning rate of 0.002, batch size of 64, and hyperparameters $\gamma = 3.0$ and $\eta = 1.0$ for 6 epochs.

Table 3 shows that Dyslexify outperforms DP on two out of three typographic benchmarks, while DP retains slightly higher performance on two out of three non-typographic benchmarks. Notably, DP yields a modest accuracy improvement on the corresponding non-typographic ImageNet-100, likely caused by the choice of ImageNet-100-typo as the training set for DP. We hypothesize that the black-box optimization in DP captures features relevant not only to typographic defense but also to ImageNet-100 classification, thereby limiting its generalization. In contrast, our method focuses on key mechanisms relevant to typographic attacks, leading to more robust transfer across non-typographic benchmarks.

Table 3: Performance comparison of Dyslexify and Defense-Prefix (DP) on ViT-L across typographic and non-typographic datasets. For each method we show the accuracy followed by the deviation from the baseline, (↑) for improvement, (↓) for decline.

| Method | Real Typographic | | | Training | Non-Typographic | |
|---|---|---|---|---|---|---|
| | RTA-100 | Disentangling | PAINT | ImageNet-100 | Food-101 | Aircraft |
| Baseline | 54.40 | 50.56 | 61.81 | 79.76 | 90.48 | 36.36 |
| DP | 62.20↑7.80 | 82.78↑32.20 | 71.82↑10.01 | 81.70↑1.94 | 89.83↓0.65 | 32.94↓3.42 |
| Dyslexify | 71.00↑16.60 | 60.56↑10.00 | 76.36↑14.55 | 79.52↓0.24 | 89.31↓1.17 | 34.62↓1.74 |

## 5.5 DEFENDING AGAINST TYPOGRAPHIC ATTACKS IN MELANOMA DETECTION

Safety-critical domains such as medicine are particularly vulnerable, as AI decisions may impact human lives. Therefore, we investigate whether typographic attacks transfer to this setting and

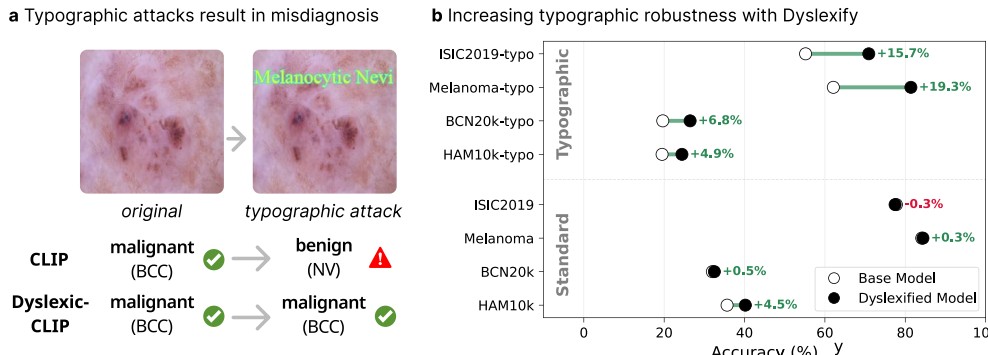

Figure 6: Typographic attacks in melanoma detection. (a) Adding adversarial text may cause CLIP to misdiagnose a malignant lesion as benign. (b) Applying Dyslexify mitigates these failures, increasing robustness to typographic attacks and even improving accuracy in several non-attacked cases.

whether our defense remains effective. Specifically, we analyze WhyLesionCLIP, a foundation model for skin lesion classification (Yang et al., 2024), i.e., melanoma detection, based on OpenClip-ViT-L.

We utilize the same setting as in Section 5.1 retrieving the typographic attention scores, but deviate from Section 5.2 in that we construct $\mathcal{C}$ by setting ISIC2019 as $D_{\text{img}}$ and ISIC2019-Typo as $D_{\text{typo}}$.

The results in Fig. 6 and in Table 4 reveal two key insights: (i) Typographic attacks reduce the zero-shot accuracy of melanoma detection by up to 22% and (ii) Dyslexify proves effective in defending a medical foundation model from a relevant attack vector. Not only does Dyslexify increase the accuracy under typographic attack by up to 19.3%, but it also increases the base models accuracy in three out of the four datasets. An additional medical use-case is analyzed in Appendix E.

## 6 CONCLUSION

We present a mechanistic defense against typographic attack in CLIP using an interpretability-first approach. We reveal that a small number of attention heads located in the later layers of the vision encoder are responsible for the effectiveness of typographic attacks. By selectively ablating a typographic circuit, Dyslexify is able to defend CLIP against typographic attacks without requiring fine-tuning steps, offering a practical and interpretable method for controlling model behavior. To our knowledge, this is the first work to address typographic attacks in CLIP through causal interventions. Dyslexify demonstrates that fine-grained control over model capabilities is achievable through targeted architectural manipulations *without* retraining, and thus paves the way for more robust and modular deployment of multimodal models. Beyond standard benchmarks, we further show that typographic attacks constitute a realistic threat vector in the medical domain, where they can mislead safety-critical models, and that Dyslexify substantially mitigates this vulnerability. We believe this work motivates a broader shift toward mechanistic interpretability as a tool not only for understanding, but for controlling safety-relevant behaviors in deep transformer models. Finally, we release a family of dyslexic CLIP models that are significantly more robust against typographic attacks. These models serve as drop-in replacements for safety-critical applications where the risks posed by adversarial text manipulation outweigh the benefits of typographic understanding.

**Limitations and future work:** Dyslexify enhances the typographic robustness of the `cls` token by preventing specialized typographic attention heads from writing to it. However, many multimodal applications, such as LLaVA and IP adapters (Ye et al., 2023; Liu et al., 2023; 2024b), leverage not only the `cls` token but also spatial tokens, allowing typographic information to propagate into downstream tasks. This might limits the impact of Dyslexify on improving robustness in applications, and calls for further investigation into its generalizability to VLM setups.

Furthermore is it standard practice to evaluate adversarial defenses against adaptive attacks (Tramer et al., 2020), i.e., attacks that are explicitly optimized to circumvent the defense mechanism. In our

case, however, such an evaluation is not feasible: typographic attacks are inherently non-differentiable, which prevents constructing adaptive variants that directly optimize against Dyslexify.

**Misuse Potential:** While we aim to enhance the safety of multimodal systems, we acknowledge that our insights into CLIP's behavior under typographic attacks could be exploited by attackers. Specifically, adversarial inputs might be crafted to increase the spatial attention of heads in $\mathcal{C}$, making typographic attacks even more effective.

## ACKNOWLEDGEMENTS

This work was supported by the German Research Foundation (DFG) as research unit `DeSBi` `[KI-FOR 5363] (459422098).`

## ETHICS STATEMENT

We affirm that this work adheres to the ICLR Code of Ethics. All authors have read and agree with the Code. We note that our study does not involve human subjects or sensitive personal data. While insights into typographic attacks may inform adversarial strategies, our primary aim is to strengthen robustness and safety of multimodal models.

## REPRODUCIBILITY STATEMENT

We have taken efforts to make our results reproducible. Our code is open-sourced and provided to reviewers in anonymized form. All results and the majority of plots in the paper can be reproduced with the released code.

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

CONTENTS OF APPENDIX

## A   USE OF LARGE LANGUAGE MODELS

Large Language Models (LLMs) were used in the preparation of this work. Specifically, they assisted in polishing the text, generating code snippets, and conducting literature searches. All conceptual contributions, experimental designs, analyses, and conclusions are the sole work of the authors.

## B   MLP LAYERS COMPRESS INFORMATION

In Section 3, we observe that $\text{Acc}(P_{\text{img},\ell})$ and $\text{Acc}(P_{\text{typo},\ell})$ increases after attention layers, but consistently drops following MLP blocks. Thus we estimate the intrinsic dimensionality (ID) of `cls`-token representations across layers using PCA. From a 5% split of the ImageNet-100-Typo training set, we extract residual stream activations for each layer. For each embedding $h_\ell \in \mathbb{R}^d$, we fit PCA and define ID as the smallest number of principal components $k$ such that the cumulative explained variance exceeds 95%:

$$\text{ID} = \min \left\{ k : \frac{\sum_{j=1}^{k} \lambda_j}{\sum_{j=1}^{d} \lambda_j} \geq 0.95 \right\},$$

where $\lambda_j$ are the PCA eigenvalues. A larger ID indicates higher representational complexity. We report ID across layers and token types to analyze how typographic inputs affect the geometry of CLIP representations.

**Results**   Fig. 7 compares linear probe accuracy (left) and ID (right) across layers of a ViT-B model. We observe a consistent pattern: attention layers tend to increase both probe accuracy and intrinsic dimensionality, while MLP layers reduce them. This trend is most prominent in the middle layers (3–7), suggesting that attention blocks introduce linearly accessible information, whereas MLPs compress or remove it.

This pattern is not consistent throughout the model. The MLPs at layers 9 and 11 exhibit increases in ID, deviating from the overall compression trend. Conversely, the attention block at layer 11 causes a sharp drop in ID. These exceptions indicate that deeper layers may serve more specialized roles.

Attention Layers add information to the CLS tokens, while MLPs remove information from it

**a)** This can be observed in the probe accuracy

**b)** And in intrinsic dimensionality analysis

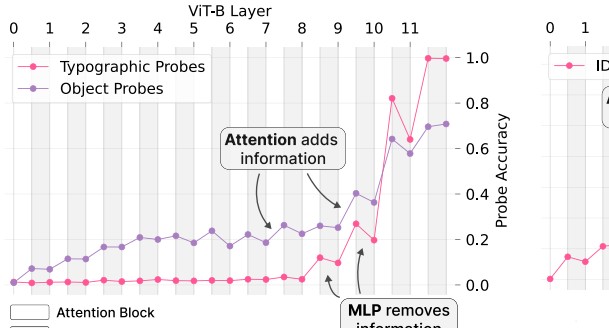
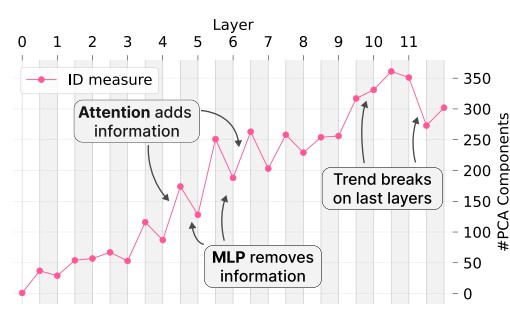

Figure 7: Attention layers increase, MLP layers reduce `cls` token information. **(a)** Linear probe accuracy rises after attention blocks and drops after MLPs, indicating improved linear accessibility followed by compression. **(b)** Intrinsic dimensionality shows a matching trend, especially in layers 3–7. Exceptions include MLPs at layers 9 and 11 (ID increase) and the attention block at layer 11 (ID drop), suggesting deeper-layer specialization.

## C  ATTENTION SINKS FOR TYPOGRAPHIC ATTENTION HEADS

We analyze the attention patterns of head 5 in layer 6 ($\mathcal{H}_{5,6}$) in ViT-B, which has the highest $T_{i,\ell}$ score in the model. More specifically we evaluate its spatial attention norm $\|A_{5,6}^*\|$ on ImageNet-100 and ImageNet-100-Typo.

As shown in Fig. 8, $\|A_{5,6}^*\|$ is systematically higher on ImageNet-100-Typo than on ImageNet-100. While the distribution on typopgrahic images is unimodal, the distribution on the original dataset is bimodal. Manual inspection reveals that the high-norm mode in ImageNet-100 contains incidental text in, such as watermarks or copyright tags.

One possible interpretation of these results is that $\mathcal{H}_{5,6}$ uses the `cls`-to-`cls` attention as an attention sink (Xiao et al., 2023), to selectively adjust the impact of this specialized typographic attention head.

Building on these finding we evaluate $\mathcal{H}_{5,6}$'s capabilities to predict if a sample $x$ originates from ImageNet-100 or ImageNet-100-typo. The score $\|A_{5,6}^*\|$ ROC-AUC of 0.887, indicating that this attention signal can be used as a robust classifier. In comparison, a linear classifier trained on the same task reaches an ROC-AUC of 1.0, but overfits to superficial typographic features specific to the ImageNet-100-Typo construction. Many images in the original dataset that contain real-world typography are still correctly classified as non-typographic by this probe - supporting the conclusion that it does not generalize beyond the synthetic intervention.

## Spatial attention norm is selective

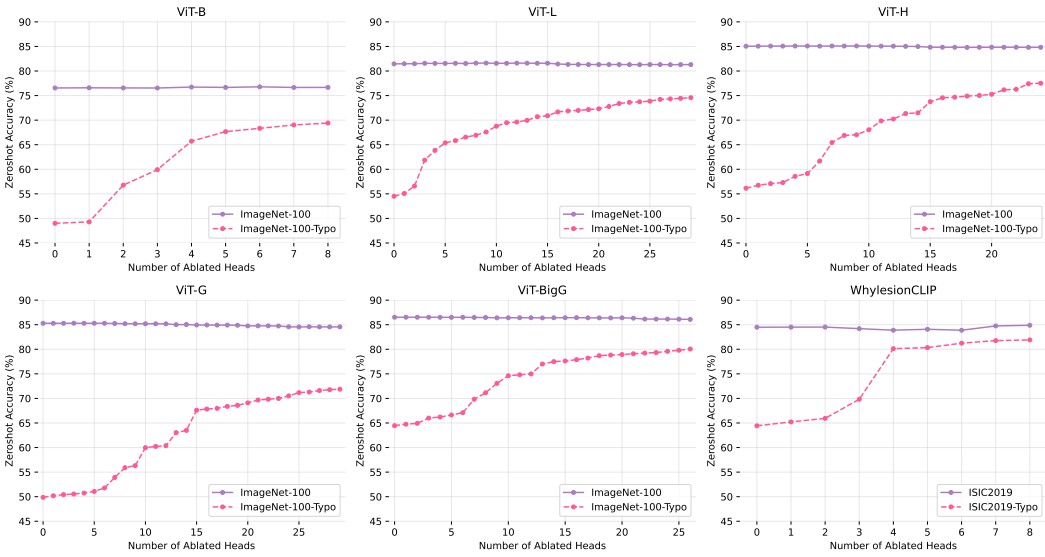

Figure 8: Distribution of the spatial attention norm of ViT-B head $\mathcal{H}_{5,6}$ across ImageNet-100 and ImageNet-100-Typo. The norm is consistently higher for typographic images, while the non-typographic distribution is bimodal. Manual inspection links the higher mode to incidental text, suggesting that the head selectively activates in response to typography, regardless of its origin.

## D  ABLATION CURVES

Figure 9: Tradeoff between general accuracy and typographic robustness as a function of the number of ablated heads. Ablations are applied in decreasing order of $T_{i,\ell}$.

# E    FURTHER DETAILS ON MEDICAL APPLICATION

We evaluated Dyslexify in the safety-critical context of melanoma detection, using WhyLesionCLIP based on OpenCLIP ViT-L as the underlying foundation model. To construct typographic attack datasets, we introduced textual labels into ISIC2019, HAM10k, and BCN20k, resulting in paired typographic variants. The typographic circuit was selected using the same procedure as in Section 5.2, with tolerance $\epsilon = 0.01$ and maximum skips $k = 10$.

Table 4 reports detailed results. We find that typographic attacks reduce zero-shot accuracy by up to 22.1%, while Dyslexify recovers up to 19.3% accuracy, and even improves performance on non-attacked datasets in some cases. These results confirm that typographic attacks pose a realistic failure mode for medical AI systems.

Fig. 10 further illustrates that typographic probes perform more strongly in WhyXrayCLIP than in WhyLesionCLIP, which may be explained by frequent typographic artifacts (e.g., "R" markers) in X-ray training data. This suggests that the degree of vulnerability depends on the presence of typographic features in the training distribution.

Table 4: Comparison of dyslexic model performance on WhylesionCLIP dataset, showing accuracy changes relative to the base model, with improvements (↑) or declines (↓). The suffix -T indicates the corresponding typographic version of the dataset.

| Model | Not Typographic | | | | Synthetic Typographic | | | |
|---|---|---|---|---|---|---|---|---|
| | ISIC2019 | Melanoma | BCN20k | HAM10k | ISIC2019-T | Melanoma-T | BCN20k-T | HAM10k-T |
| Base | 77.80 | 84.10 | 31.99 | 35.66 | 55.19 | 62.10 | 19.65 | 19.48 |
| Ours | 77.47 ↓0.33 | 84.40 ↑0.30 | 32.49 ↑0.50 | 40.20 ↑4.54 | 70.92 ↑15.73 | 81.40 ↑19.30 | 26.46 ↑6.81 | 24.42 ↑4.94 |

**a)** Typographic probes perform more strongly in the lesion model than in the xray model

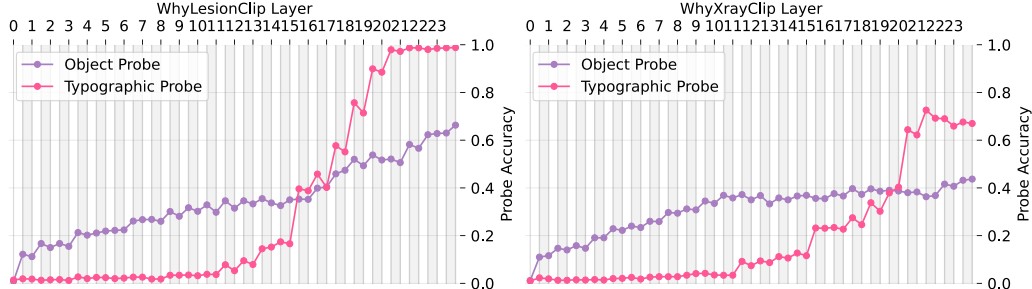

**b)** Typographic artifacts in training data (e.g., R-markers) may explain robustness

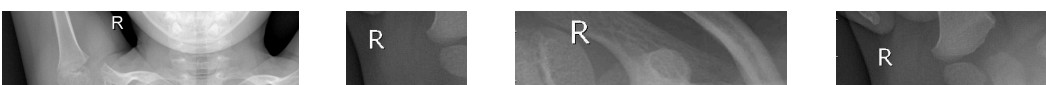

Figure 10: (a) Linear probes on CLS embeddings show that typographic probes perform less well in WhyLesionCLIP compared to WhyXrayCLIP, indicating weaker encoding of typographic features in the lesion model. (b) Examples of typographic artifacts (e.g., "R" markers) commonly found in X-ray training data, which may influence probe performance.

## F    DATASETS

For the zero-shot evaluation we tested a variety of datasets, grouped below by purpose.

**RTA-100** (Azuma & Matsui, 2023) consists of 1,000 handcrafted typographic attacks, each written on a Post-it note and overlaid onto natural images.

**Disentangling** (Materzyńska et al., 2022) includes 180 typographic attacks, also written on Post-its, designed to probe the separation of visual and textual features in multimodal models.

**PAINT-DS** (Ilharco et al., 2022) comprises 110 Post-it-based typographic attacks and serves to evaluate patch-level vulnerabilities in vision-language models.

**Food-101** (Bossard et al., 2014) is a standard image classification dataset containing 101 food categories, each with 1,000 images.

**FGVC-Aircraft** (Maji et al., 2013) (referred to as *Aircraft* in our paper) is a fine-grained classification benchmark consisting of 10,000 images across 100 aircraft variants.

**ImageNet-100**[2] is a subset of the ImageNet-1k dataset (Deng et al., 2009), containing 100 object and animal classes with 1,000 images per class.

**ISIC2019**[3] contains 25,331 images available for the classification of dermoscopic images among nine different diagnostic Tschandl et al. (2018); Combalia et al. (2019); Codella et al. (2018). In this paper we evaluate the binary classification task into the classes "Benign" and "Malignant".

**HAM10k** (Tschandl et al., 2018) includes 10,015 dermatoscopic images labeled into seven diagnostic categories: *Actinic Keratoses*, *Basal Cell Carcinoma*, *Benign Keratosis-like Lesions*, *Dermatofibroma*, *Melanoma*, *Melanocytic Nevi*, and *Vascular Lesions*.

**BCN20k** (Combalia et al., 2019) comprises 19,424 dermatoscopic images collected at the Hospital Clínic de Barcelona, annotated into eight categories: *Actinic Keratoses*, *Basal Cell Carcinoma*, *Benign Keratosis-like Lesions*, *Dermatofibroma*, *Melanocytic Nevi*, *Melanoma*, *Squamous Cell Carcinoma*, and *Vascular Lesions*.

For the generation of the -typo dataset we used the following fonts: Times New Roman, Georgia, Arial.

## G    CIRCUIT SIZE

Table 5: Number of selected and total heads $\mathcal{H}_{i,\ell}$ in circuit $\mathcal{C}$ per model

| Model | Selected | Total | Percentage (%) |
|-------|----------|-------|----------------|
| B     | 8        | 144   | 5.6            |
| L     | 29       | 288   | 10.1           |
| H     | 24       | 384   | 6.2            |
| G     | 29       | 480   | 6.0            |
| Big-G | 26       | 576   | 4.5            |

## H    OCR CAPABILITIES

Dyslexify is designed to suppress typographic understanding in favor of robustness to typographic attacks. As a consequence, we expect a degradation on tasks that require optical character recognition (OCR).

To quantify this effect, we evaluate the base and dyslexic OpenCLIP models on the IIIT5K word recognition benchmark (Mishra et al., 2012), using the standard small and medium lexicon settings.

---

[2] https://www.kaggle.com/datasets/ambityga/imagenet100
[3] https://www.kaggle.com/datasets/salviohexia/isic-2019-skin-lesion-images-for-classification

For each image, we treat the lexicon entries as candidate labels and perform zero-shot classification with CLIP, reporting the top-1 word accuracy.

Table 6 reports the results. Across all model sizes, Dyslexify substantially reduces OCR performance, with drops between 8 and 30 percentage points depending on the model and lexicon size. The effect is strongest for the smaller ViT-B and ViT-L models and remains pronounced even for the larger ViT-bigG.

This confirms that dyslexic models are not appropriate for applications that benefit from text recognition. Instead, they are intended as drop-in replacements in safety-critical settings where the risks of adversarial text manipulation outweigh the utility of typographic understanding.

Table 6: Comparison of dyslexic model performance on IIIT5K OCR evaluation, showing accuracy changes relative to the base model, with improvements ($\uparrow$) or declines ($\downarrow$).

| Model | Small Lexicon | Medium Lexicon |
|-------|---------------|----------------|
| B | $25.03_{\downarrow 30.50}$ | $11.37_{\downarrow 17.87}$ |
| L | $35.50_{\downarrow 30.27}$ | $15.30_{\downarrow 20.47}$ |
| H | $51.27_{\downarrow 12.23}$ | $26.40_{\downarrow 8.57}$ |
| G | $51.40_{\downarrow 13.70}$ | $27.27_{\downarrow 10.03}$ |
| Big-G | $54.47_{\downarrow 20.37}$ | $29.40_{\downarrow 15.67}$ |

## I  RUNTIME AND MEMORY REQUIREMENTS

We benchmark the computational cost of Dyslexify against Defense-Prefix (DP) in terms of wall-clock runtime and memory requirements. All experiments were run on a single NVIDIA H200 GPU using identical data pipelines and batch sizes. Importantly, both methods were evaluated using the *entire ImageNet-100 training split* for circuit construction and prefix optimization, respectively.

Table 7: Runtime comparison (in seconds) of Dyslexify and Defense-Prefix (DP) in seconds on a single NVIDIA H200 GPU. Lower is better.

| Model | Dyslexify | DP |
|-------|-----------|-----|
| ViT-B | 3 818 | 14 598 |
| ViT-L | 23 518 | 23 145 |
| ViT-H | 24 237 | 39 483 |

Dyslexify is $3.8\times$ faster than DP on ViT-B, $1.6\times$ faster on ViT-H, and comparable on ViT-L. The cost gap reflects the fact that Dyslexify requires only forward passes with circuit ablations, whereas DP performs gradient-based optimization over a learnable prefix.

**Runtime reduction through subset evaluation.**  The reported Dyslexify runtimes use the *full* ImageNet-100 training set. In practice, Dyslexify does not require the entire dataset: we observe that using a significantly smaller random subset yields nearly identical circuits and comparable robustness gains.

**VRAM requirements.**  Dyslexify further has a low memory footprint. It runs on all evaluated models, including those exceeding 1B parameters, on a single NVIDIA Titan RTX (24 GB). Defense-Prefix cannot be trained beyond ViT-L due to CUDA out-of-memory errors. This makes Dyslexify suitable for low-resource environments where finetuning-based defenses are infeasible.

## J  BLURRING BASELINE

Following the reviewer's suggestion, we implemented a simple OCR-based defense baseline that removes typographic content directly from the image. We use the PaddleOCR text detector (Cui et al.,

2025) to localize text regions and then apply a Gaussian blur to each detected bounding box using OpenCV (Bradski, 2000)..

In brief, for each input image we (1) run an OCR pass to obtain polygonal text boxes; (2) compute the pixel bounds of each box; and (3) blur only the corresponding subregion with a large-kernel Gaussian filter (kernel size 51, $\sigma = 25$). No other pixels are modified. This produces a text-ablated variant of ImageNet-100-Typo that preserves the underlying image content while suppressing overlaid words.

This baseline allows direct comparison between Dyslexify (a mechanistic intervention) and a purely image-level text-removal defense.

The OCR+blur baseline is strong: it consistently outperforms Dyslexify on ImageNet-100-Typo across all model sizes. This confirms that removing text at the pixel level is an effective strategy for mitigating typographic attacks. On non-typographic datasets, the blur baseline has minimal impact - the accuracy remains within $\pm 0.3\%$ of the original models - indicating that the image modifications do not substantially harm object-centric features. Dyslexify shows similarly small degradation (<1%), but delivers lower absolute robustness than OCR+blur on this benchmark.

While OCR-based preprocessing is competitive in absolute robustness, it comes with clear limitations. First, OCR+blur must be applied at every inference step, adding recurring computational overhead, whereas Dyslexify is executed once per model and introduces no inference-time cost. Second, OCR+blur removes visual information rather than modifying internal causal pathways; it therefore does not provide mechanistic insight into model behavior or contribute to understanding the underlying failure modes.

Overall, the results highlight that OCR-based preprocessing is a strong baseline, but Dyslexify serves a different purpose: it demonstrates that mechanistic analysis can yield actionable interventions that improve robustness without retraining and without modifying the input image.

We hope that this comparison highlights a broader point: while preprocessing-based defenses can be strong, they do not replace the need for mechanistic approaches. Our results show that even simple circuit-level interventions can yield meaningful robustness gains without retraining, without modifying inputs, and without inference-time overhead. We hope that this work encourages further research on actionable interpretability - using mechanistic understanding not only to analyze models, but to directly improve their safety and reliability.

Table 8: Comparison of dyslexic model performance on ImageNet-100 with and without blur.

| ImageNet-100 | B | L | H | G | Big-G |
|---|---|---|---|---|---|
| Normal | 74.36 | **79.76** | 83.74 | **83.24** | **85.06** |
| Blurred | 74.32 | 79.48 | **83.88** | 83.20 | 84.94 |
| Dyslexify | **75.00** | 79.52 | 83.40 | 82.58 | 84.72 |
| Both | 74.64 | 79.50 | 83.50 | 82.52 | 84.60 |

Table 9: Comparison of dyslexic model performance on ImageNet-100-Typo with and without blur.

| ImageNet-100-Typo | B | L | H | G | Big-G |
|---|---|---|---|---|---|
| Normal | 46.94 | 51.90 | 54.08 | 46.70 | 61.90 |
| Blurred | **68.52** | **75.12** | 79.98 | **79.56** | **80.84** |
| Dyslexify | 66.84 | 72.22 | 75.34 | 68.76 | 78.64 |
| Both | 67.86 | 74.30 | **80.00** | 78.48 | 80.52 |

## K RELATIONSHIP BETWEEN PROBES AND $T_{i,\ell}$ SCORES

As discussed in Section 3, we observe a strong correspondence between layers with elevated $T_{i,\ell}$ scores and those where the typographic probe $P_{\text{typo},\ell}$ exhibits sharp increases in accuracy. In this section, we extend this analysis across all evaluated model sizes to support the trends previously

shown for ViT-B. Fig. 11 to Fig. 15 visualize this relationship for each model. To improve readability, we transpose the probe accuracy plots: the $y$-axis denotes the layer index, while the $x$-axis indicates probe accuracy.

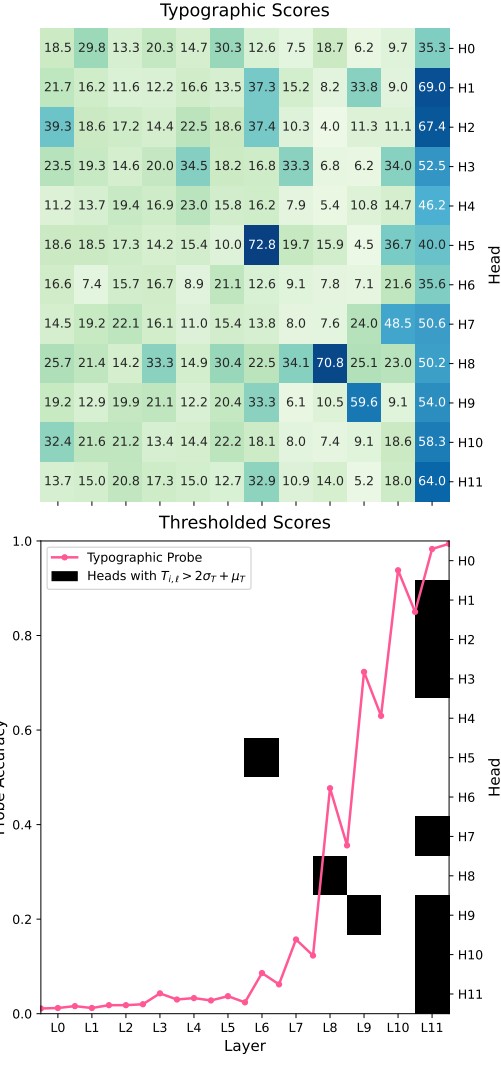

Figure 11: ViT-B typographic attention scores $T_{i,\ell}$ (top), thresholded and compared with linear probe accuracies (bottom). Layers with higher typographic attention scores align with increased probe accuracy, highlighting a correspondence between attention patterns and typographic feature decodability.

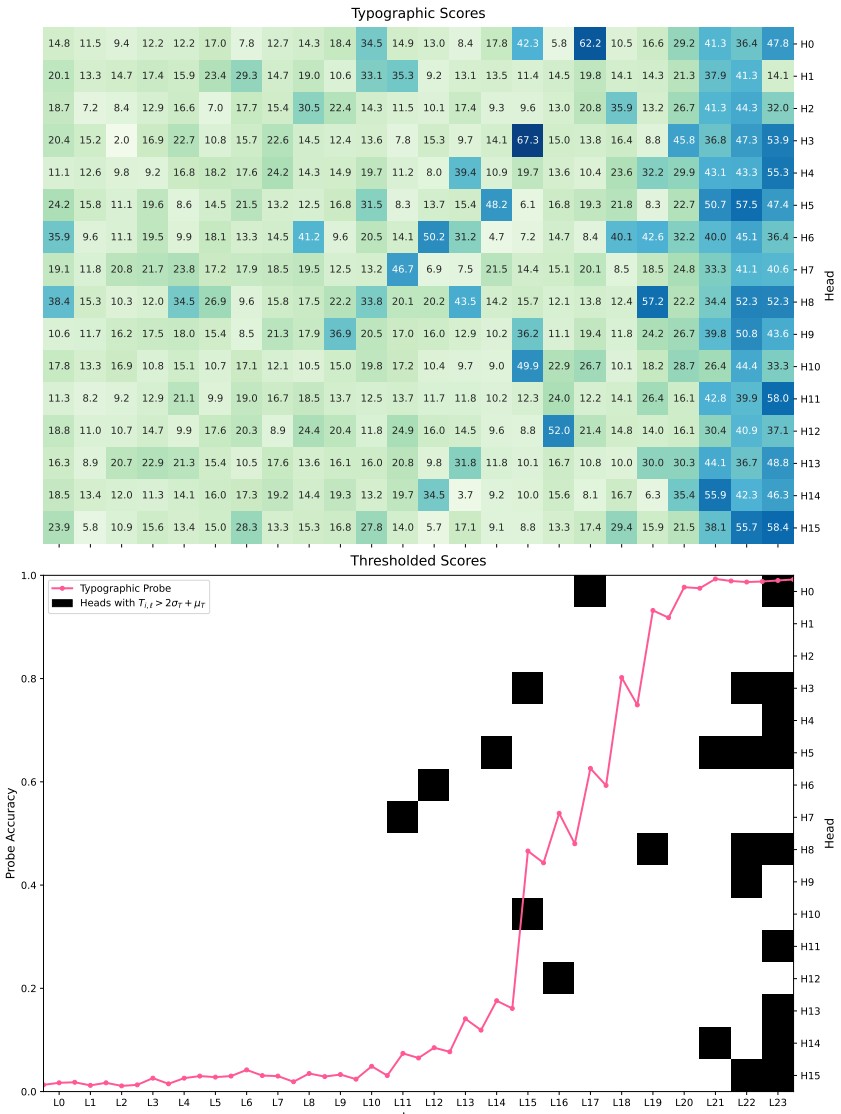

Figure 12: ViT-L typographic attention scores $T_{i,\ell}$ (top), thresholded and compared with linear probe accuracies (bottom). Layers with higher typographic attention scores align with increased probe accuracy, highlighting a correspondence between attention patterns and typographic feature decodability.

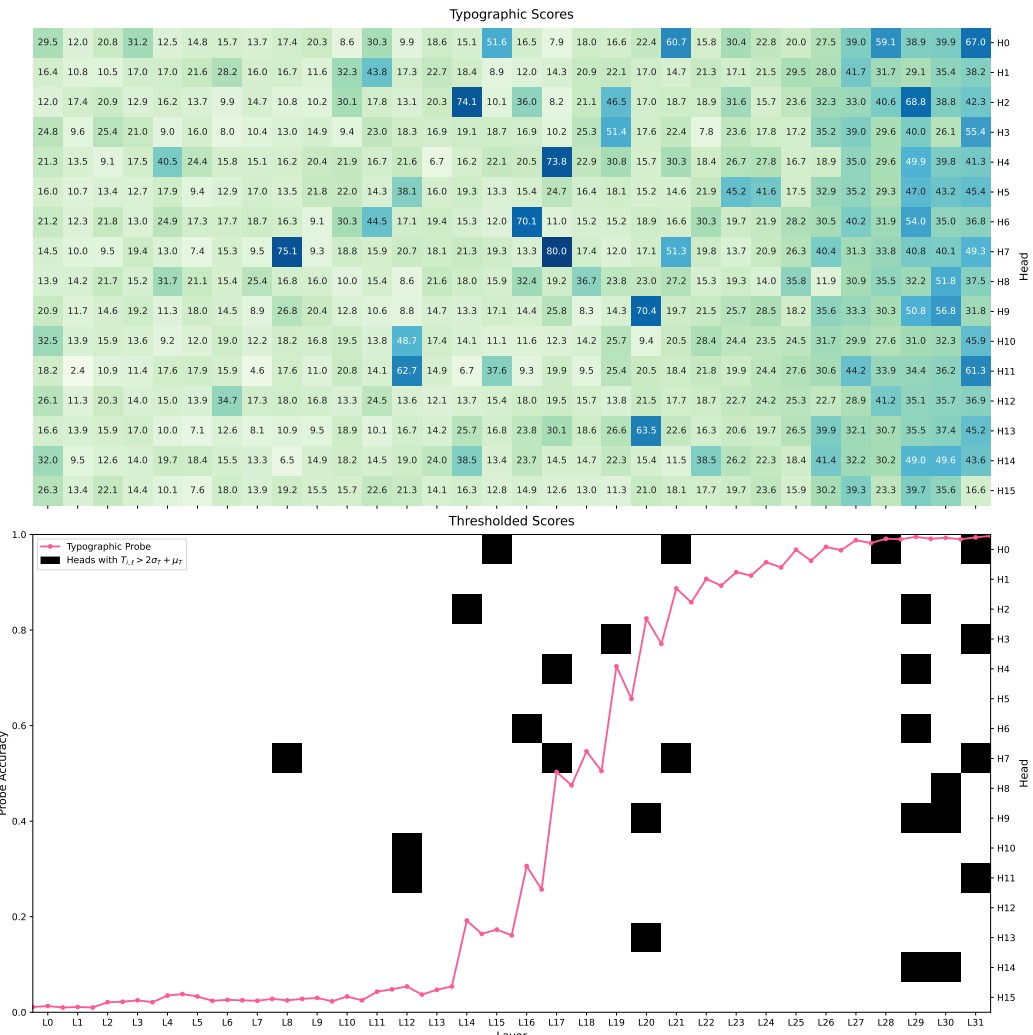

Figure 13: ViT-H typographic attention scores $T_{i,\ell}$ (top), thresholded and compared with linear probe accuracies (bottom). Layers with higher typographic attention scores align with increased probe accuracy, highlighting a correspondence between attention patterns and typographic feature decodability.

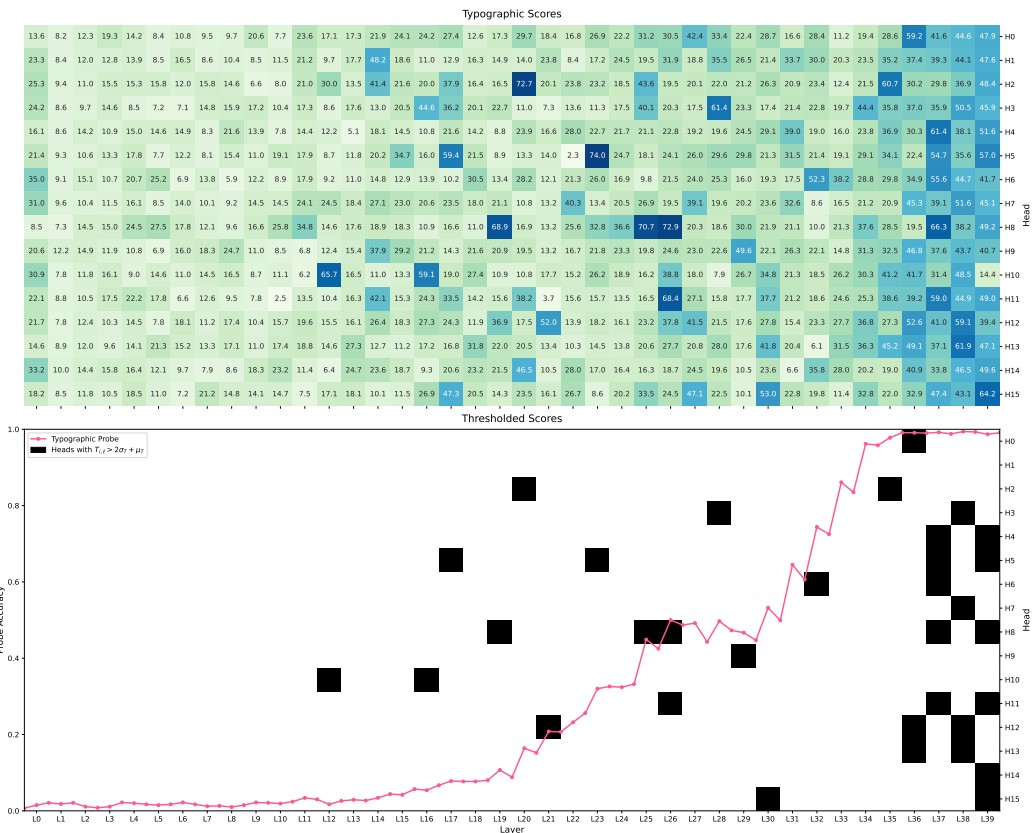

Figure 14: ViT-G typographic attention scores $T_{i,\ell}$ (top), thresholded and compared with linear probe accuracies (bottom). Layers with higher typographic attention scores align with increased probe accuracy, highlighting a correspondence between attention patterns and typographic feature decodability.

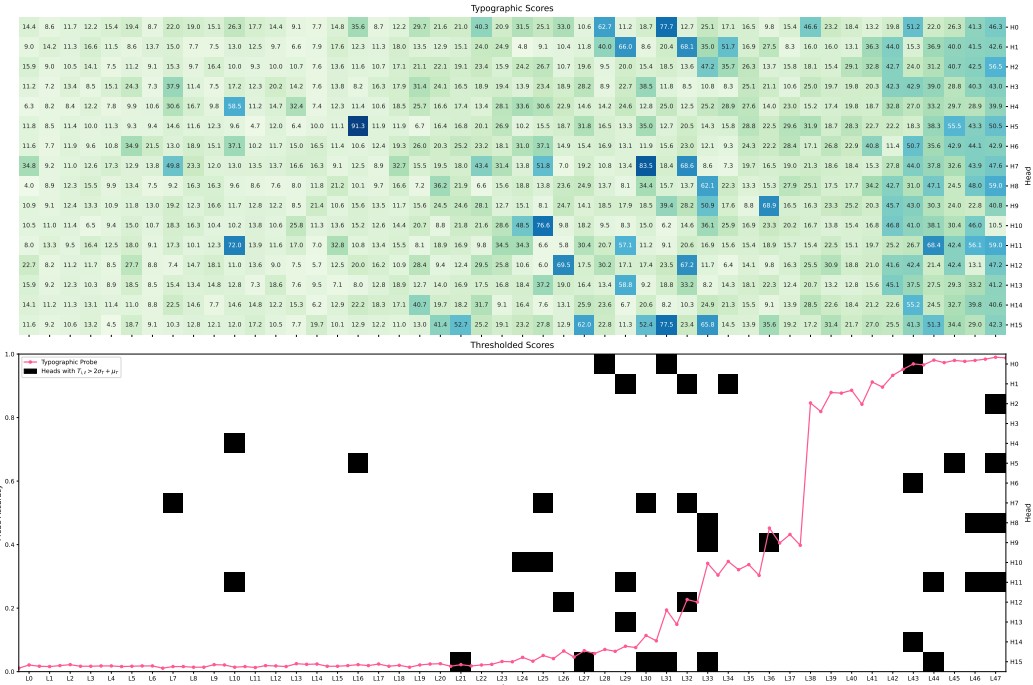

Figure 15: ViT-BigG typographic attention scores $T_{i,\ell}$ (top), thresholded and compared with linear probe accuracies (bottom). Layers with higher typographic attention scores align with increased probe accuracy, highlighting a correspondence between attention patterns and typographic feature decodability.

