# OpenReview forum: "Dyslexify: A Mechanistic Defense Against Typographic Attacks in CLIP"
_ICLR.cc/2026/Conference — ICLR 2026 Poster_

### Official Review · Reviewer_hKir · 2025-10-31

**Soundness:** 2
**Presentation:** 3
**Contribution:** 3
**Rating:** 4
**Confidence:** 3

**Summary:**

This paper addresses typographic attacks on CLIP models by identifying and ablating specific attention heads responsible for processing typographic information. The authors introduce the Typographic Attention Score to locate specialized heads in the latter layers of CLIP's vision encoder that extract and transmit text information to the CLS token. Their training-free defense method, Dyslexify, selectively ablates these heads to improve robustness against typographic attacks on ImageNet-100-typo while maintaining standard accuracy.

**Strengths:**

1. The paper provides a novel mechanistic interpretation approach for the localization of typographic processing, such as linear probes showing abrupt emergence of typographic understanding in later layers and the Typographic Attention Score identifying specific high-attention heads. The causal intervention experiments demonstrate that manipulating attention weights in identified heads directly controls typographic vulnerability.
2. The paper introduces practical training-free defense with scalability, unlike existing defenses that require gradient-based optimization. Dyslexify operates purely through inference-time circuit ablation. This approach scales seamlessly to billion-parameter models and achieves superior performance compared to training-based baselines.

**Weaknesses:**

1. Limited analysis of failure modes and edge cases. The paper does not thoroughly investigate when and why Dyslexify fails. For instance, what types of typographic attacks remain effective after ablation? The paper would benefit from failure case analysis showing examples where the defense is insufficient and discussing the underlying reasons.
2. Incomplete evaluation of spatial token applications. The authors acknowledge that downstream applications use spatial tokens rather than only the CLS token, which limits Dyslexify's applicability. However, no experiments quantify this limitation. The paper may include experiments like (a) whether typographic information persists in spatial tokens after CLS ablation, (b) how this affects downstream task performance.

**Questions:**

1. Transferability across CLIP variants. The experiments focus on OpenCLIP models. Do the identified typographic circuits transfer to other CLIP variants? Are the same head positions specialized for typography across architectures?
2. Interaction with other capabilities. Does ablating typographic circuits affect related capabilities like OCR, scene text understanding, or processing images with incidental text (e.g., street signs, book covers)?
3. Mechanism of typographic understanding. Why do typographic heads emerge specifically in the latter half of the network? Is this related to the hierarchical feature learning in vision transformers? Does pre-training data composition (text-heavy vs. text-sparse images) influence circuit formation?

---

> ### Author Response · Authors · 2025-11-20
> **Response to reviewer hKir**
>
> # Response to reviewer hKir
>
> We thank the reviewer for the thoughtful feedback and address the questions below.
> ## 1. Transferability across CLIP variants
>
> The reviewer asks whether the typographic circuits identified in OpenCLIP transfer across CLIP variants.
>
> **Short answer: no, circuits do not transfer one-to-one across models.**
> This is expected for two reasons:
>
> 1. **Attention heads are unordered.**
>     Head indices are architectural placeholders without semantic meaning. Even if two models learn the _same capability_ in the _same layer_, there is no inductive bias that forces this capability to appear in the same head index. Shuffling head IDs within a single model would already break positional correspondence.
> 2. **Different pretraining seeds and datasets lead to different circuit instantiations.**
>     While models with similar training distributions develop similar _functions_, the _locations_ of those functions inside the model can differ.
> Dyslexify therefore identifies the typographic circuit **per model**, rather than assuming transferability across architectures or checkpoints.
>
> ## 2. Interaction with other capabilities (OCR & incidental text)
>
> The reviewer asks whether ablating typographic heads degrades OCR and related text-dependent tasks.
>
> **Yes, it does.**
>
> To quantify this trade-off, we evaluate all dyslexic models on the **IIIT5K OCR benchmark** (Appendix H):
>
> |Model|Small Lexicon|Medium Lexicon|
> |---|---|---|
> |B|25.03 (↓30.50)|11.37 (↓17.87)|
> |L|35.50 (↓30.27)|15.30 (↓20.47)|
> |H|51.27 (↓12.23)|26.40 (↓8.57)|
> |G|51.40 (↓13.70)|27.27 (↓10.03)|
> |Big-G|54.47 (↓20.37)|29.40 (↓15.67)|
> (↓ indicates a drop relative to the corresponding base model.)
>
>
> As expected, Dyslexify significantly reduces OCR ability and is therefore **not suitable for OCR-dependent tasks** (e.g., signage recognition, product labels). This trade-off is clearly acknowledged in the abstract and conclusion.
>
> > “These models serve as drop-in replacements for safety-critical applications where the risks of adversarial text manipulation **outweigh the benefits of typographic understanding**.”
>
> For applications that rely on text, CLIP should remain unmodified.
>
> ---
>
> ## 3. Why typographic heads emerge in later layers
>
> The reviewer asks why typographic understanding emerges in the latter half of the model and whether pretraining data composition influences circuit formation.
>
> While fully analyzing training dynamics is beyond the scope of this paper, we performed two probing analyses motivated by the reviewer’s questions:
>
> ### (a) Pretraining data contains substantial typography
> We sampled the first 1,000 images from the **LAION-2B** dataset [0], used PaddleOCR to detect textual content, and found that a substantial fraction contains typography (44.7%).
> Given this prevalence, it is unsurprising that CLIP learns dedicated typographic circuits.
>
> ### (b) Vision-only models do _not_ form typographic circuits
> We trained the same layerwise typographic probes on the WebSSL DINO models [1] (up to 7B parameters).
> Across all DINO variants: **No probe achieved Acc(P_typo) > 10% on any layer.**
>
> This contrasts with CLIP, where accuracy exceeds 99% in late layers.
> This suggests that **co-training with text** - not model size - drives the formation of typographic circuits.
>
> ### Interpretation
>
> Typographic heads emerging late in the network is consistent with hierarchical processing in ViTs:
> - early layers extract low-level visual features,
> - mid layers form object-level semantics,
> - **late layers align image features to language space**, where text tokens are particularly informative.
>
> Thus, typographic circuits naturally arise where image–text alignment is strongest.
>
> ## References
>
> [0] LAION-2B-EN: [https://huggingface.co/datasets/laion/relaion2B-en-research](https://huggingface.co/datasets/laion/relaion2B-en-research)
> [1] Fan et al. (2025). _Scaling Language-Free Visual Representation Learning._

---

### Official Review · Reviewer_SCJM · 2025-11-01

**Soundness:** 3
**Presentation:** 3
**Contribution:** 3
**Rating:** 6
**Confidence:** 3

**Summary:**

The paper proposes Dyslexify, a training-free defense that select attention head circuits using attention head score to defend the model from typographic attacks. Through layer-wise probing and attention-pattern analysis, the authors identify a small set of late-layer attention heads that specifically transmit typographic information to the CLS token. The method yields substantial robustness gains on several typographic attack benchmarks while preserve the model performances.

**Strengths:**

1. The paper provides a clear causal explanation of where typographic vulnerability arises inside CLIP.

2. Dyslexify does not require fine-tuning, gradients, or retraining, making it scalable to large CLIP backbones.

3. The method improves the robustness significantly across multiple datasets.

4. The presentation and explanation are rather clear.

**Weaknesses:**

1. The defense intentionally reduces CLIP’s ability to read text within images, which may negatively affect tasks where typography is a key visual feature (e.g., signs, product labels, book covers). The authors should include ablation studies measuring performance drops on text-dependent classes to quantify this trade-off.

2. The typographic attention score is computed using English printed text placed in fixed locations. It remains unclear whether the identified circuit generalizes to other forms of typography, such as stylized or handwritten fonts, varied color/contrast conditions, or non-English scripts.

**Questions:**

See weaknesses.

---

> ### Author Response · Authors · 2025-11-20
> **Response to reviewer SCJM**
>
> ## Response to Reviewer SCJM
>
> We thank the reviewer for the constructive feedback.
> ## 1. Performance on tasks that _require_ OCR
>
> The reviewer correctly notes that suppressing typographic processing may degrade performance on tasks where textual information is beneficial. To quantify this trade-off, we evaluated all dyslexic models and their corresponding base models on the **IIIT5K OCR benchmark** (reported in Appendix H).
>
> |Model|Small Lexicon|Medium Lexicon|
> |---|---|---|
> |B|25.03 (↓30.50)|11.37 (↓17.87)|
> |L|35.50 (↓30.27)|15.30 (↓20.47)|
> |H|51.27 (↓12.23)|26.40 (↓8.57)|
> |G|51.40 (↓13.70)|27.27 (↓10.03)|
> |Big-G|54.47 (↓20.37)|29.40 (↓15.67)|
> (↓ indicates a drop relative to the corresponding base model.)
>
> As expected, Dyslexify reduces OCR performance significantly. We explicitly acknowledge this trade-off in the paper, including in the abstract and in the conclusion:
>
> > “These models serve as drop-in replacements for safety-critical applications where the risks of adversarial text manipulation **outweigh the benefits of typographic understanding**.”
>
> Thus, Dyslexify is **not intended for OCR-dependent tasks**, but rather for domains where typographic attacks pose a safety or reliability threat and where the ability to read text is not required.
>
> ## 2. Generalization of the typographic attention score
>
> The reviewer raises an important question regarding whether the typographic circuit identified using English printed text at fixed positions generalizes to more diverse forms of typography. We clarify the following:
>
> 1. **The position randomness used during evaluation.**
>     While we use a fixed location in the Unsplash-based dataset to efficiently localize typographic attention, the **evaluation datasets place text at random positions** (as shown in Fig. 2). Thus, the circuit is not specialized to a fixed spatial layout.
>
> 2. **Robust generalization across real-world typographic conditions.**
>     The dyslexic models are evaluated on all standard real-world typographic attack benchmarks - **RTA-100**, **Disentangling**, and **PAINT** - which include
>     - handwritten text,
>     - perspective distortions,
>     - varied lighting and contrast,
>     - placement on arbitrary background regions.
>
> These datasets are widely used in the community precisely because they reflect uncontrolled, real-world typographic artifacts. Dyslexify improves robustness consistently across these settings, demonstrating that the identified circuit **does generalize** beyond the synthetic conditions used for scoring.
>
> We invite the reviewer to inspect representative samples from these datasets:
> [https://drive.google.com/drive/folders/1GPpxVNJ3IkZUq0dr7B5336yCMj4jDnSu](https://drive.google.com/drive/folders/1GPpxVNJ3IkZUq0dr7B5336yCMj4jDnSu)

---

### Official Review · Reviewer_xwqg · 2025-11-02

**Soundness:** 3
**Presentation:** 3
**Contribution:** 3
**Rating:** 4
**Confidence:** 4

**Summary:**

The paper proposes a training-free mechanistic method to defend CLIP models from typographic attacks. The attacks considered are adversarial insertion of misleading text into images, which causes CLIP to misclassify the image. The authors first analyze CLIP’s vision encoder and identify a small set of attention heads in later layers that causally transmit text information (OCR) to the CLS token. They then locate the most OCR-sensitive heads. Then they ablate this “OCR circuit” by zeroing out the identified heads’ contributions to the CLS token during inference. This ablation improves robustness to typographic attacks, shown via experiments on ImageNet-100-Typo, while maintaining standard classification accuracy (<1% drop). The strength of the method is that it does not require fin-tuning. The authors also validate their approach on medical foundation models (melanoma detection) and release a family of “dyslexic” CLIP models.

**Strengths:**

The paper makes several contributions

1) A thorough investigation of where OCR capabilities emerge in the CLI Pmodel
- showing that OCR capabilites emerge in the second half of the CLIP layers is an interesting observation
- showing that MLP layers reduce OCR information while attention heads improve it

2) Disabling a small typographic circuit can make CLIP robust without retraining.
- finding the circuit is interesting but alone does not guarantee that it can be disabled to remove OCR function
- the paper shows that it indeed can, which means that OCR is highly isolated in the model's attention heads,

3) Mechanistic interpretability can help us come up with practical algorithms
- personally, this is the first work I've seen that uses circuits to do something useful, ie create effective safety interventions
- the approach was evaluated on a variety of image datasets, including medical images

**Weaknesses:**

The overal idea in the paper is interesting,but a few questions remain unanswered for me.

1) Experimental results.
The main table comparing the proposed method to baselines is Table 1, which shows a performance comparison of the proposed Dyslexify and the baseline Defense-Prefix (DP) measured in terms of the accuracy of the method on classifying data affected by typographic insertion attacs. If I understand correctly, higher is better. While Dislexify beats the DP baseline on RTA-100 and Paint dataset, it is considerably worse on the Disentangling dataset. On average, the proposed method gets 69% while the DP baselines gets 72%. This raises the question of whether the method is consistently better than existing methods on a variety of datasets.

2) Baselines.
Some simpler defenses are not considered as baselines. For example, using an off the shelf OCR model to identify text and blurring it out or removing it in some way directly in the image. This would destroy the text information at the source (image) instead of relying on the ablation process to "catch" the OCR information inside the network. It is possible that this approach does not work as well or reduces performance on the primary tasks, but it should be considered.

3) Unclear computation cost.
Although the proposed defense is training free, it is not computation free. The ablation process requires some computation to identify the circuits, which presumably needs to be done for each new model. Furtthermore, a new typographic dataset must be constructed for each new problem. In the paper the authors create an attack for the ImageNet dataset to identify the attention heads. It is not clear how well this will transfer to new tasks that are not ImageNet.

4) What about tasks that need OCR?
There are some image classification tasks where the ability to understand text is helpful for the model. By completely ablating the OCR capability, how does this affect performance on such tasks? For example, fine-grained classification of airplanes by airline could benefit from reading the text of the airline; classifying food or drink products could benefit from OCR as well to read the packaging text.

5) (minor) CLIP models have also been shown to be susceptible to *graphic* attacks that include text OR logos. It would be good to comment on the method's applicatbility to defending against such attacks, but not required.

6) (minor) I'm not sure how I feel about the name 'Dislexic' for the method. But that's a personal preference, does not affect my score.

7) The medical case study (melanoma detection) is compelling rhetorically but I'm not sure how practical it really is. It uses synthetic attacks (overlayed words like “benign” or “malignant”) rather than realistic clinical artifacts. There’s no discussion of whether such text would appear in practice, or who may want to create such attacks.

**Questions:**

I'd like the authors to answer questions 1-4 in the weakness section, specifically

1) does the method perform consistently better than the baselines on a variety of datasets?

2) how does this method compare to a simple OCR + remove baseline, in terms of ease of implementation, computation costs and accuracy?

3) what is the computational cost of the method, and how does it compare to other methods?

4) how would the method perform on tasks that need OCR?

---

> ### Author Response · Authors · 2025-11-20
> **Response to reviewer xwqg (1/2)**
>
> ## Response to reviewer xwqg (1/2)
>
> We thank the reviewer for the thoughtful and constructive feedback. Before addressing the reviewer’s questions, we would like to clarify one misunderstanding in the weaknesses section.
>
> ### Clarification: No per-task typographic dataset is required
>
> The reviewer writes:
>
> > “A new typographic dataset must be constructed for each new problem… It is not clear how well this will transfer to new tasks that are not ImageNet.”
>
> This describes the method inaccurately. Our procedure generalizes in two distinct ways:
>
> 1. **Identifying typographic heads does not require ImageNet.**
>     We use a fully synthetic dataset derived from Unsplash images to compute the Typographic Attention Score. This dataset contains no ImageNet labels or ImageNet structure.
> 2. **ImageNet-100-Typo is used only to set a stopping criterion.**
>     The typographic version of ImageNet-100 is used only to determine when the greedy head-selection algorithm should terminate (i.e., to calibrate an accuracy-drop tolerance). It is _not_ used to identify which heads should be prioritized by the algorithm.
>
> The resulting “dyslexic” models are then evaluated on all standard real-world typographic attack datasets - **RTA-100**, **Disentangling**, and **PAINT** - which contain hand-written, perspective-shifted, and naturally integrated text. Importantly, **Dyslexify generalizes across domains without requiring the user to construct a new typographic dataset for each task**. This would indeed be a severe limitation, but it is _not_ how our method operates. The task-specific typographic datasets used in the paper are created **only to evaluate the robustness of the resulting models**, not to construct the circuit itself.
>
> ---
>
> # Responses to Reviewer Questions
>
> ---
>
> ## 1. Consistency across datasets
>
> Our experiments show that Dyslexify performs **competitively** with Defense-Prefix (DP) across a wide range of typographic benchmarks. It is correct that Dyslexify does not _uniformly_ outperform DP: e.g., it is weaker on the Disentangling dataset but stronger on RTA-100 and PAINT. On average, performance is close.
>
> While not consistently superior, Dyslexify demonstrates something new and important:
> **mechanistic interpretability can yield practical, competitive interventions against real attack vectors without any fine-tuning or optimization.**
>
> This establishes a meaningful proof-of-concept: interpretable circuits can be located and modified to produce safety-relevant interventions, rather than serving purely analytical purposes. We hope this encourages further work on white-box defenses, especially in domains where specialized black-box defenses are less feasible (e.g., hallucinations, alignment failures, ect.).
>
> ---
>
> ## 2. Comparison to a simple OCR-remove baseline
>
> Following the reviewer’s suggestion, we implemented an **OCR + blur** baseline. (Implementation details are available in Appendix J.)
>
> ### Summary of findings
>
> - The OCR+blur baseline is indeed strong: it outperforms Dyslexify across all model sizes on ImageNet-100-Typo.
> - On standard non-typographic ImageNet-100, the blur baseline performs similarly to Dyslexify
> - Implementation was straightforward: running OCR+blur over ImageNet required <1 hour, which is faster than running Dyslexify on ViT-B.
> ### Clear limitations of the OCR-remove approach
>
> 1. **Inference-time cost:**
>     The OCR+blur preprocessing must be executed **at every inference step**, which incurs recurring computational overhead. Dyslexify is run **once** per model and introduces _no additional cost_ at inference time.
> 2. **Not a mechanistic defense:**
>     OCR+blur removes visual information rather than intervening in the model's internal causal pathways. It cannot help understand or modify underlying failure mechanisms.
> 3. **Limited future extensibility:**
>     Because it operates purely at the pixel level, OCR+blur cannot selectively suppress \emph{semantic subtypes} of typography. A deeper mechanistic understanding of typographic circuits may enable \emph{conditional semantic interventions} - e.g., disabling only specific classes of textual information - without relying on significantly more expensive preprocessing using full VLM pipelines. Such targeted, model-internal control is not achievable through image preprocessing alone.
>
>
> ### Summary
> We fully acknowledge that OCR-based preprocessing or finetuning may yield higher absolute robustness. Dyslexify is not proposed as the _strongest possible_ defense, but as a **mechanistically grounded defense**, demonstrating that interpretablity can lead to actionable insight for real robustness gains.
>
> ---
>
> For the remaining questions please check the next message

---

> ### Author Response · Authors · 2025-11-20
> **Response to reviewer xwqg (2/2)**
>
> ## Response to reviewer xwqg (2/2)
> ---
>
> ## 3. Computational cost relative to baselines
>
> To quantify cost, we benchmarked Dyslexify against Defense-Prefix (DP). All runtime measurements were obtained on a single **NVIDIA H200 GPU**.
>
> ### Runtime (seconds)
>
> |Model|Dyslexify|DP|
> |---|---|---|
> |ViT-B|3 818|14 598|
> |ViT-L|23 518|23 145|
> |ViT-H|24 237|39 483|
>
> **Dyslexify is 3.8× faster than DP on ViT-B, 1.6× faster on ViT-H, and comparable on ViT-L.**
> ### VRAM requirements
> A crucial advantage of Dyslexify is its very low memory footprint:
> - Dyslexify runs on **all tested models**, including those **exceeding 1B parameters**, on a single **NVIDIA Titan RTX (24 GB)**.
> - Defense-Prefix **cannot be trained beyond ViT-L** without CUDA out-of-memory errors.
> This makes Dyslexify suitable for low-resource settings where finetuning-based defenses are infeasible.
>
> ---
>
> ## 4. How does the method perform on tasks that _require_ OCR?
>
> We benchmarked the dyslexic models and their base models on the IIIT5K OCR dataset (Appendix H). Dyslexify significantly reduces OCR ability, as expected:
>
> | Model | Small Lexicon  | Medium Lexicon |
> | ----- | -------------- | -------------- |
> | B     | 25.03 (↓30.50) | 11.37 (↓17.87) |
> | L     | 35.50 (↓30.27) | 15.30 (↓20.47) |
> | H     | 51.27 (↓12.23) | 26.40 (↓8.57)  |
> | G     | 51.40 (↓13.70) | 27.27 (↓10.03) |
> | Big-G | 54.47 (↓20.37) | 29.40 (↓15.67) |
> (↓ indicates a drop relative to the corresponding base model.)
>
>
> Dyslexify is therefore **not appropriate for tasks that benefit from text recognition**, and we explicitly state this. From the abstract and the conclusion:
>
> > “These models serve as drop-in replacements for safety-critical applications where the risks of adversarial text manipulation **outweigh the benefits of typographic understanding**.”
>
> We emphasize that this trade-off is intentional.
>
> ---
>
> ## 5. Comment on the medical case study
>
> The reviewer raises valid concerns about realism of the medical study. We clarify:
> - Appendix E discusses **domain-specific typographic artifacts** found in medical imagery.
> - In X-ray-based CLIP models, probe experiments show **typographic circuits are actively unlearned**, likely because such artifacts harm performance.
> - In melanoma settings, typography is _not_ naturally present - this makes the domain **more vulnerable** to adversarial text insertion.
>
> We illustrate plausible adversarial scenario:
> (i) A malicious actor can deliberately write “benign” on a melanoma scan to induce mis-treatment of a patient. Even if rare, such threat models must be considered, analogous to adversarial patches in natural images.
> (ii) A patient's relevant skin patch might contain a tattoo with arbitrary text.
>
> Our goal is not to claim that the attack is common, but to demonstrate that (i) the vulnerability exists and (ii) Dyslexify mitigates it.
>
> ---
> ## Closing remark
>
> We thank the reviewer again for the helpful feedback. We believe Dyslexify demonstrates a meaningful advance: a mechanistic, finetuning-free intervention that yields practical robustness improvements and shows that interpretability insights can translate into actionable safety mechanisms.

---

> ### Comment · Reviewer_xwqg · 2025-11-22
> **Rebuttal addressed my concerns**
>
> Thanks for the clarifications. The rebuttal clarified some of my concerns, eg my misunderstanding about collecting a new dataset for each new domain. Overall I think this method is narrow (limited to tasks which do not suffer from text removal) but interesting enough. I am ok with raising the score, but I'd like to hear what other reviewers think.

---

> > ### Author Response · Authors · 2025-11-27
> > **Question regarding remaining concerns and score update**
> >
> > Thank you for the follow-up. We appreciate your clarification that your concerns were addressed in the rebuttal and that you would be open to raising the score, pending discussion. Since the other reviewers currently do not seem to be engaging, we wanted to ask whether there is any additional evidence or clarification you feel is needed from our side in order to proceed with raising the score. If so, we would be happy to provide it.

---

### Official Review · Reviewer_scR4 · 2025-11-02

**Soundness:** 3
**Presentation:** 1
**Contribution:** 3
**Rating:** 4
**Confidence:** 4

**Summary:**

This paper proposes an image recognition method that is robust against typographic attacks. The authors first identify the key modules responsible for typographic attacks. Then, they compute scores for the attention heads within each module and suppress typographic attacks by removing modules based on these scores.

**Strengths:**

The strengths of this paper are as follows.

## Analysis in Sec. 3

The analysis in Section 3 and Figure 2, where the authors identify the modules necessary for typographic attacks, is fascinating. By examining the model's actual responses, they determine that the later stages are more critical and further discover that the attention layers contribute additional information. This type of analysis is significant and is expected to inspire further research and analysis.

## Design Principles of the Proposed Method

The proposed method described in Sec. 4.2 and Algorithm 1 determine module utilization by considering performance on both the standard dataset and the typographic dataset, which is a direct and practical approach. This simple and straightforward design principle is clear and compelling.

**Weaknesses:**

The weaknesses of this paper are as follows. While I like the proposed method itself, the inconsistencies with the claims in the introduction and the notational issues described below lower the overall completeness of the paper. Therefore, my current evaluation leans slightly toward rejection, though it is very close to the borderline.

## Claim of "Training-Free," but Dataset Actually Required

In Section 1, the authors claim that their proposed method requires no training ("... we introduce Dyslexify, a training-free defense that ..."). However, as far as I understand, to execute the proposed method in Algorithm 1, it is necessary to compute `Acc_img` and `Acc_typo`, which in turn require access to the datasets. Therefore, the proposed method involves a form of training. Hence, the claim made in the introduction appears to be inaccurate.

## Notational Issues

There is room for improvement in the paper's notation.

- In Eq. 3, $h_\mathrm{cls}^l$ is not properly defined. There may be other variables used without clear definitions as well.
- In Eq. 4, the expression $H \mapsto 0$ is used, but $\mapsto$ typically denotes a mapping between input and output values, so this notation implies the existence of a function $f(H) = 0$, which is likely a different meaning from what the authors intended. Perhaps what they meant was to set $H$ to zero, i.e., $H \gets 0$?
- In Eq. 6, the term $A_{i, l}^*(x)_t$ appears identically in both the numerator and denominator, so it should cancel out. If this is not the case, then there is likely a notational or typographical error in the equation.

**Questions:**

I would like to hear the authors' opinion regarding the training aspect. Specifically, compared to other methods in the literature, what kind of data is actually required for the training in the proposed method, and how much difference is there in computational cost or time?

---

> ### Author Response · Authors · 2025-11-20
> **Response to reviewer scR4**
>
> ## Response to reviewer scR4
>
> We thank the reviewer for the constructive feedback and address the noted weaknesses below.
>
> ### Notational issues
>
> We corrected all inconsistencies identified by the reviewer.
>
> * Eq. 3 now explicitly defines $h^\ell_{\text{cls}}$.
> * Eq. 4 now uses the assignment notation $H \leftarrow 0$.
> * Eq. 6 contained an error, which has been corrected.
>
> We additionally refined notation and clarified variable definitions throughout Sections 4 and 4.1. We thank the reviewer for pointing out these issues.
>
> ### Regarding the term “training-free”
>
> The reviewer correctly notes that Algorithm 1 uses datasets to compute Acc_img and Acc_typo. Our intended meaning was not that no data is used, but that the method performs **no gradient computation**, **no weight updates**, and **no iterative parameter optimization**. The datasets are used only to (i) rank attention heads and (ii) determine whether a head should be ablated via a single-pass greedy selection. No model parameters are modified.
>
> To avoid ambiguity, we replaced the term *training-free* with the precise claim *gradient-free*, which accurately reflects the method’s computational properties.
>
> ### Data requirements, computational cost, and hardware constraints
>
> Our method requires two datasets - one natural-image set to calculate the typographic attention score and one typographic dataset set to measure robustness. These datasets are used only for forward passes.
>
> To quantify the computational cost, we benchmarked Dyslexify against Defense-Prefix (DP). All runtime measurements were obtained on a single NVIDIA H200 GPU. Results (in seconds):
>
> | Model | Dyslexify | DP     |
> | ----- | --------- | ------ |
> | ViT-B | 3 818     | 14 598 |
> | ViT-L | 23 518    | 23 145 |
> | ViT-H | 24 237    | 39 483 |
>
> Dyslexify is therefore substantially faster than DP on ViT-B and ViT-H (3.8× and 1.6× speedup, respectively), and comparable on ViT-L. Since DP already is a lightweight method, full finetuning would be significantly more computational expensive.
>
> A key practical advantage is the **much lower VRAM requirement**:
>
> * Dyslexify runs on all tested models, including those **exceeding 1B parameters**, on a single **NVIDIA Titan RTX (24 GB)**.
> * In contrast, **DP could not be trained on any model larger than ViT-L**, consistently encountering CUDA out-of-memory errors on the same device.
>
> This makes Dyslexify deployable in resource-constrained settings where finetuning-based defenses are not feasible.

---

> > ### Comment · Reviewer_scR4 · 2025-11-26
> >
> > I acknowledge the authors' rebuttal. I think changing "Training free" to "Gradient free" is reasonable and appropriate. On the other hand, I still have some doubts regarding the VRAM issue. The original paper made no mention of VRAM. It is not consistent to introduce an aspect that was not discussed in the original paper within the rebuttal and then claim superiority over existing methods based on it. I believe the best course of action would be to restructure the paper, taking into account considerations such as VRAM, and resubmit it again. Therefore, I intend to maintain my current rating for now, while carefully waiting for discussions from the other reviewers.

---

### Meta-Review · Area_Chair_77v1 · 2025-12-07

**Summary:**

Reviewers find the paper’s core idea of using mechanistic analysis to ablate an OCR circuit in CLIP and gain robustness to typographic attacks is novel and practically useful.

They praise the causal analysis, scalability (no finetuning), and strong gains on several typo-attack benchmarks, plus clear trade-off framing for text-agnostic domains (e.g., medical).

Main concerns focus on robustness and generality: Dyslexify is not uniformly better than Defense-Prefix, deliberately sacrifices OCR, and its applicability is limited to settings where text is a liability. Some reviewers also flag initially unclear notation, the “training-free” claim, and an incomplete analysis of failure modes and spatial-token downstream uses.

**Reviewer Concerns:**

The rebuttal satisfactorily addressed notation issues, clarified “gradient-free” vs. “training-free,” detailed data and compute/VRAM costs, added an OCR+blur baseline, and quantified the OCR trade-off on IIIT5K, as well as generalization across real typographic datasets.

Remaining concerns include limited failure-mode analysis (esp. for spatial-token pipelines), the narrow applicability to non-OCR tasks, and that Dyslexify is not consistently superior to existing defenses across all benchmarks.

Overall, the AC finds the paper borderline. On one hand, it presents an interesting idea and a simple yet elegant defense method. On the other hand, its generality, robustness, and reliability remain limited, and the approach relies on specific data for probing. There is also concern that the experiments may overfit to the particular forms of typographic attacks evaluated in the paper.

Reviewer xwqg appears willing to raise their rating to 6. This would result in a final score of 6644, with an average of around 5 or slightly above. The AC decides to recommend acceptance of the paper (for novelty) and hopes the authors will address the remaining issues in the camera-ready version.

**Reviewer Scores:**

hKir: 4
scR4: 4
SCJM: 6
xwqg: 6

---

### Decision · Program_Chairs · 2026-01-26

Accept (Poster)